# Modeling and interpreting hydrological responses of sustainable urban drainage systems with explainable machine learning methods

Yang Yang[1], Ting Fong May Chui[1]

[1]Department of Civil Engineering, The University of Hong Kong, Hong Kong SAR, China

*Correspondence to*: Ting Fong May Chui (maychui@hku.hk)

**Abstract.** Sustainable urban drainage systems (SuDS) are decentralized stormwater management practices that mimic natural drainage processes. The hydrological processes of SuDS are often modeled using process-based models; however, the application of these models can be challenged by insufficient data regarding the physical properties and drainage processes of the catchment. To address this problem, this study proposes a machine learning method to directly learn the statistical

correlations between the hydrological responses of SuDS and the preceding rainfall time series collected at sub-hourly time scales from observation data. A feature engineering method is developed to facilitate the application of machine learning algorithms to model high-dimensional time series data. A local feature attribution method is used to explain the basis of each discharge prediction, i.e., the contribution of rainfall of each time step is identified for each prediction. These local explanations are then combined to investigate the correlations between the input and output variables learned by the models and infer the

modeled hydrological processes. The proposed methods are applied to two SuDS catchments with different sizes, SuDS practice types, and data availabilities in the U.S. for discharge prediction. The resulting models have high prediction accuracies (the Nash–Sutcliffe model efficiency coefficient (NSE) > 0.70 and the coefficient of determination ($R^2$) > 0.70 for all models). The local explanations of the predictions are analyzed and the results show that the discharge predictions are mostly affected by rainfall from the most recent time steps, and that the rainfall–discharge correlations learned by some models are not realistic.

These explanations are further used to decompose hydrographs and identify the runoff source. This study shows that machine learning methods are a useful tool for predicting the hydrological responses of SuDS, and the explanation methods can provide insights on the structures and input–output correlations learned by the models, both of which enhance the understanding of the modeled catchment processes.

## 1 Introduction

Sustainable urban drainage systems (SuDS), also known as low impact development (LID) practices, green infrastructure (GI), and sponge city, are decentralized stormwater management practices that aim to promote onsite infiltration, storage, evapotranspiration, and stormwater reuse (Fletcher et al., 2015; Jones and Macdonald, 2007). SuDS can effectively improve stormwater runoff quality, reduce runoff volume, and restore natural hydrological regimes (Selbig et al., 2019; Trinh and Chui, 2013; Zhou, 2014). Commonly used SuDS practices include bioretention cells, green roofs, porous pavement, and rain barrels

(Charlesworth, 2010; Gimenez-Maranges et al., 2020). It is essential to be able to predict the hydrological performance of SuDS and understand the involved hydrological processes to support their design optimization, planning, and related policy making (Pappalardo and La Rosa, 2020; Yang and Chui, 2018).

A number of numerical modeling methods of various complexities have been developed for SuDS (Elliott and Trowsdale, 2007; Liu et al., 2014). The simplest methods are perhaps those developed based on empirical equations for computing the
drainage impact of different land use types in terms of peak flow or runoff volume. For instance, the rational method and SCS runoff curve number method are modified and used in Montalto et al. (2007) and Damodaram et al. (2010) to study the effectiveness of SuDS at catchment scales. Empirical equation-based methods can be useful in preliminary designs to rapidly estimate some key performance metrics of SuDS. However, these methods poorly reflect detailed SuDS design variations. For example, it is often unclear how to vary the curve number in the SCS method for different green roof substrate depths (Fassman-
Beck et al., 2016). The relatively coarse temporal and spatial resolutions of empirical equation-based methods can also limit their application.

Process-based models are another approach to modeling SuDS, in which physically based or empirical equations are used to characterize the involved hydrological processes. SuDS are typically represented in process-based models as hydrological functional units, whose properties are defined using a set of parameters. Commonly used models, including SWMM and
MUSIC, are reviewed in Eckart et al. (2017) and Elliott and Trowsdale (2007). A process-based model can ideally be set up for a catchment using only its measured physical properties (Refsgaard and Storm, 1990). However, not all parameters are measurable or can be measured at a reasonable cost. Model calibrations are often required and critical for SuDS (Platz et al., 2020). For example, in the SWMM, the initial soil moisture deficit parameter of SuDS is often determined through calibration (Rosa et al., 2015). However, model calibration can be difficult and subject to considerable uncertainty (Schuol and Abbaspour,
50 2006).

The application of process-based models faces several challenges. First, it can be difficult to choose a suitable model to represent SuDS. SuDS designs can substantially vary in terms of their installation location, layer composition, and materials. Unfortunately, models that are capable of modeling all of these design variants may not exist. Modular-form models, such as GIFMod, represent SuDS as a collection of hydrological functional components arranged in 1D or 2D grids, which allows
different design variants to be modeled (Massoudieh et al., 2017). However, these models are arguably more complex and do not necessarily produce more accurate predictions despite the higher data requirements for the model setup. Second, the complex hydro-environmental processes of SuDS and surrounding environments are difficult to model using existing models. For instance, the SWMM does not account for macropore flow in the SuDS soil layer (Niazi et al., 2017), and models that assess the performance of SuDS in shallow groundwater environments (Zhang and Chui, 2019) and cold climates (Johannessen
et al., 2017) are limited. Third, the assumptions used in process-based models may be invalid in some cases due to unknown issues related to construction, maintenance, or physical property changes during a SuDS service life. For instance, the permeability of porous pavement changes over time due to factors such as clogging (Yong et al., 2013), but the processes that contribute to clogging and its associated impact on infiltration processes remain poorly understood. Complex environment

processes or human errors, such as changes in ecological processes (Levin and Mehring, 2015) and inferior constructions, are often unknown to modelers and thus difficult to predict. Other model types that rely less on assumptions are therefore preferred under certain circumstances.

Machine learning methods, also referred to as data-driven modeling, predictive modeling, and statistical learning, may be used to learn the statistical correlations between the SuDS state variables of interest from observed data (Solomatine and Ostfeld, 2008). Machine learning methods have been widely used in various fields in hydrology (Maier and Dandy, 2000), such as rainfall–runoff modeling (Worland et al., 2018), evapotranspiration forecasting (Karbasi, 2018), and groundwater modeling (Jang et al., 2020).

Machine learning methods have only been used in a few studies to investigate the hydrological processes of SuDS. For instance, multiple linear regression methods were used in Eric et al. (2015) and Khan et al. (2013) to predict the hydrological effectiveness of SuDS, such as runoff volume reduction, based on factors such as inflow volume, antecedent soil moisture content, and SuDS implementation levels. Eric et al. (2015) concluded that the linear regression approach is useful for predicting the hydrological performance of SuDS in large catchments with thousands of lots and potentially eliminates the need to build complex process-based catchment models. Li et al. (2019) used neural networks models to predict the peak flow and runoff volume resulting from a rainfall event on a university campus with SuDS implementations based on rainfall event characteristics, such as rainfall depth and event duration. Hopkins et al. (2020) used linear regression models to predict the timing and volume of runoff resulting from a rainfall event in urban catchments with and without SuDS implementations. The studies mentioned above focused on predicting the long-term or rainfall event-level hydrological performance of SuDS. However, there is currently insufficient literature on the application of machine learning methods to model the hydrological responses of SuDS at regular time steps, e.g., daily, hourly, or sub-hourly. Yang and Chui (2019) showed that machine learning methods, such as deep learning methods and random forest methods, are useful for predicting the runoff response of SuDS at sub-hourly time scales, provided that the model's input variables are appropriate. However, a method to derive these input variables was not described in their study.

The lack of popularity of machine learning models in SuDS-related studies may be explained by several factors. First, the hydrological responses of SuDS are controlled by relatively long-term hydrometeorological conditions. Thus, modeling the responses of SuDS at fine temporal scales requires a high-dimensional hydrometeorological time series to be used as input, which is difficult for machine learning algorithms that are not specifically designed for modeling data sequences (Nielsen, 2019). Machine learning methods may also not offer clear advantages over equation-based methods when applied to study the performance of SuDS at the rainfall event level. It is therefore desirable to use a machine learning method that can handle high-dimensional time series data to model the hydrological responses of SuDS at sub-hourly time scales.

Machine learning models are also often criticized for a lack of transparency because the basis for each prediction can be difficult to analyze (Solomatine et al., 2008). Such information is important for model diagnosis, understanding the involved processes, and supporting decision making (Lundberg and Lee, 2017). To explain the basis of model predictions in hydrology, previous studies have analyzed the rules used in model trees for making predictions (Solomatine and Dulal, 2003), examined

the temporal evolution of hidden values in neural networks (Kratzert et al., 2019), referred to similar past events using instance-based learning (Wani et al., 2017), and used permutation feature importance methods to assess the importance of the predictors in the machine learning models (Schmidt et al., 2020). However, many of the explanation methods used in previous studies were ad-hoc and therefore not easily transferred to other studies. Most of the explanation methods used in the current hydrological literature are also global interpretation methods, which are useful for understanding general model-learned relationships (Murdoch et al., 2019). For instance, these methods can be applied to compute feature importance scores, identify important interactions between features, and assess the statistical significance of predictors (Murdoch et al., 2019). However, in cases where the hydrological processes of SuDS are modeled at sub-hourly time scales, it may be useful to explain the basis of each prediction to better understand the involved processes and increase the transparency of the machine learning approaches. Local feature attribution methods may be useful for this task because they can provide explanations regarding why a specific prediction is made by assessing the contribution of each feature to the prediction (Sundararajan and Najmi, 2020).

There is a limited amount of publicly available data concerning the hydrological processes of SuDS because they represent relatively new technologies and monitoring is often conducted by the local authorities and other interested parties. A lack of data is also common for other data types that are useful in urban hydrological studies, such as the soil moisture content and soil temperature (Schaffitel et al., 2020). It may be therefore useful to demonstrate the applications of machine learning methods to predict SuDS discharge based on preceding rainfalls because rainfall and discharge are the most commonly monitored features in SuDS sites and several rainfall–discharge datasets are available online (e.g., the United States Geological Survey Water Data for the Nation, https://waterdata.usgs.gov).

This study aims to evaluate the usefulness of explainable machine learning methods in modeling and interpreting the hydrological responses of SuDS to rainfall at sub-hourly time scales. Specific model training methods are developed to facilitate the modeling of high-dimensional time series data. The prediction accuracies of the resulting machine learning models are compared with the commonly used linear regression models and process-based hydrological models. The basis of each discharge prediction is explained using local feature attribution methods and attributed to the rainfalls from different time steps using a method proposed in this study. The outcome is applied to analyze the model structure and hydrological processes being modeled.

## 2 Methods and materials

### 2.1 Methods for interpreting machine learning models

#### 2.1.1 Local and global methods

Let the real-valued function $f: \mathbb{R}^N \to \mathbb{R}$ be the machine learning model to be explained. The input variable of $f$ is $\mathbf{X} \coloneqq (X_1, X_2, \ldots, X_N)$, which is an $N$-dimensional vector of numerical features. Let $\mathbf{x} \coloneqq (x_1, x_2, \ldots, x_N)$ denote a sample of $\mathbf{X}$. For a

particular **x**, *local feature attribution methods* aim to quantify the contribution of each $x_i$ to the model prediction $f(\mathbf{x})$ (Janzing et al., 2019). In contrast, *global interpretation methods*, also known as dataset-level interpretation methods, aim to understand the overall structure of $f$ that leads to different predictions when **x** assumes different values (Ahmad et al., 2018; Guidotti et al., 2019). More information on the various interpretation methods can be found in Molnar (2021).

### 2.1.2 Shapley values and their applications in feature attribution

The problem of distributing payoffs among players has been extensively studied in game theory. In coalitional games with transferable payoffs, coalitions (i.e., cooperative groups) can be formed by individual players and can generate payoffs to be distributed among their members. Shapley (2016) showed that a unique payoff profile exists for every coalitional game that satisfies a set of desirable axioms, which makes the payoff assignment "fair." Desirable axioms are discussed in detail later in the paper. The function that maps a coalitional game to the desired payoff profile is known as the *Shapley value*. For a coalitional game of $M$ players, the payoff assigned to player $i$ by the Shapley value, $\emptyset_i$, is computed as

$$\emptyset_i = \frac{1}{M!}\sum_{R\in\mathcal{R}}[v(S^R \cup i) - v(S^R)], \tag{1}$$

where $R$ is a permutation of the $M$ players, which specifies the order that each player joins the coalition, $\mathcal{R}$ is the space of all player permutations, $S^R$ is the set of players that join the coalition prior to player $i$, and $v: S \in \mathcal{P}(M) \to \mathbb{R}$ is a set function that maps every non-empty subset of $M$ players (i.e., each member of power set $\mathcal{P}(M)$ that is not empty) to a real number, where $v$ gives the total payoffs generated by a coalition. Therefore, $\emptyset_i$ can be interpreted as the expected marginal contribution of player $i$ (i.e., $v(S^R \cup i) - v(S^R)$) to coalitions that are randomly formed by sequentially selecting players. More information on the Shapley value can be found in Osborne and Rubinstein (1994).

There are considerable similarities between the tasks of attributing $f(\mathbf{x})$ to each $x_i$ and distributing payoffs among players in coalitional games. If a feature attribution task is formulated as a coalitional game, then each $x_i$ is regarded as a player and its contribution to $f(\mathbf{x})$ can be fairly determined using the Shapley value. However, $f$ cannot be used as $v$ in Eq. (1) because it is not defined for the subsets of $\{x_1, x_2, \ldots, x_N\}$. Lundberg and Lee (2017) thus defined $v$ using the *observational conditional expectation*, which is the expected value of $f(\boldsymbol{X})$ when the feature values of $\boldsymbol{X}$ in $S$ are known, as in

$$v(S) = E[f(\boldsymbol{X})|\boldsymbol{X}_S = \mathbf{x}_S], \tag{2}$$

Janzing et al. (2019) and Lundberg et al. (2020) defined $v$ using the *interventional conditional expectation*

$$v(S) = E[f(\boldsymbol{X})|do(S)], \tag{3}$$

where $do(S)$ represents an intervention that sets the feature values of $\boldsymbol{X}$ in $S$ to $\mathbf{x}_S$. The difference between the two expectations is that the observational expectation is the expected value of $f(\boldsymbol{X})$ conditioning on $\boldsymbol{X}_S = \mathbf{x}_S$, and the interventional expectation is the expected value of $f(\boldsymbol{X})$ when $\boldsymbol{X}_S$ is set to $\mathbf{x}_S$, and the remaining feature values not in $S$ are taken from $\boldsymbol{X}$ without conditioning on $\boldsymbol{X}_S = \mathbf{x}_S$.

After $v$ is defined, Eq. (1) can be used to compute $\emptyset_i(f, \mathbf{x})$, which can be considered as the contribution of $x_i$ to $f(\mathbf{x})$. This feature attribution method was proposed by Lundberg and Lee (2017) and is termed SHAP (SHapley Additive exPlanations). The two definitions of $v$ in Eqs. 2 and 3 result in two versions of $\emptyset_i(f, \mathbf{x})$, which are termed observational and interventional SHAP values, respectively. Lundberg and Lee (2017) showed that many local feature attribution methods, such as LIME (Ribeiro et al., 2016) and DeepLIFT (Shrikumar et al., 2017), compute feature attributions that are approximations of the SHAP values.

Both the observational and interventional SHAP values satisfy a set of desirable axioms, which is ensured by the properties of the Shapley values (Lundberg et al., 2020). (1) Local accuracy: the sum of $\emptyset_i(f, \mathbf{x})$ matches $f(\mathbf{x})$, which is given as

$$f(\mathbf{x}) = \emptyset_0(f) + \sum_{i=1}^{N} \emptyset_i(f, \mathbf{x}), \tag{4}$$

where $\emptyset_0(f)$ is $v(\emptyset)$, i.e., the expected value of $f(\mathbf{X})$. (2) Consistency: the SHAP value of a feature does not decrease when this feature has the same or a larger impact on the prediction upon a change of the model. (3) Missingness: $\emptyset_i(f, \mathbf{x}) = 0$ if the inclusion of $x_i$ to any collation that it is not a member of does not change the expected prediction value, i.e., $v(S \cup i) = v(S)$ for any $S$ that does not contain $i$. The local accuracy property ensures that the SHAP values are in the same space as the prediction, and the consistency property allows a comparison of the SHAP values obtained for the different models. The missingness property enforces that the features with no effect on the expected prediction value will have a SHAP value of 0.

However, Janzing et al. (2019) showed that when $v$ is defined using Eq. (2), the expected prediction value can be affected by the features not used by the model. This results in a non-zero observational SHAP value for the unused features. Janzing et al. (2019) recommended the use of interventional SHAP values because they do not present issues related to unused features. However, Chen et al. (2020) suggested that both observational and interventional SHAP values are useful. They claimed that the observational SHAP values are "true to the data" because they are effective in identifying the true correlations between the features and the modeled outcome, whereas the interventional SHAP values are "true to the model" because they do not credit the features that are unused by the model. The observational SHAP value of $x_i$ generally measures the value of knowing $x_i$ to predict the outcome, and the interventional SHAP value of $x_i$ corresponds to the expected changes in the model prediction when the feature $X_i$ is set to the $x_i$. Both the observational and interventional SHAP values are used in this study and the results obtained using the two methods are compared.

## 2.2 Training and testing machine learning models and applications of interpretation methods

### 2.2.1 Modeling hydrological responses of SuDS using machine learning methods

Let $Y_t$ denote the hydrological response of a SuDS site at time step $t$ and $\mathbf{X}_t$ denote the time series of the hydrometeorological conditions and other factors measured on and before time step $t$. The hydrological responses of SuDS at sub-hourly time scales are interested in this study because they are consistent with the typical temporal resolutions chosen for SuDS monitoring. $Y_t$ can be represented as a function of $\mathbf{X}_t$, given as

$$Y_t = f(\boldsymbol{X_t}), \tag{5}$$

where $\boldsymbol{X_t}$ is a random vector of multiple elements,

$$\boldsymbol{X_t} \coloneqq [P_{t-0}, P_{t-1}, P_{t-2}, \dots, E_1, E_2, \dots, E_k], \tag{6}$$

where $P_{t-i}$ is the rainfall depth recorded at time step $t-i$ for the preceding time step and $E_1$–$E_k$ represent $k$ measurements of the other antecedent hydrometeorological factors. The goal of machine learning is to approximate $f$ based on the observed samples of the input–output pairs $(x_t, y_t)$.

### 2.2.2 Feature engineering methods

The hydrological responses of SuDS can be affected by relatively long-term hydrometeorological conditions that occurred in the past. $\boldsymbol{X_t}$ can thus be a long time series of hydrometeorological condition measurements. As pointed out by Nielsen (2019), many machine learning algorithms are not designed for modeling time series data. Therefore, long time series are often converted into lower-dimensional features that are then used as the input variables of machine learning models. The input variable transformation process is known as feature engineering and is expected to produce higher-quality models (Kuhn and Johnson, 2019). The task of machine learning then becomes finding a function $g$ that predicts $Y_t$ based on features $\varphi(\boldsymbol{X_t})$,

$$Y_t = g\big(\varphi(\boldsymbol{X_t})\big), \tag{7}$$

where $\varphi$ is an arbitrarily defined function that transforms $\boldsymbol{X_t}$ into lower-dimensional features or features that are useful for predicting $Y_t$. Previous studies often treated $\varphi$ as a function that extracts the summary statistics from a rainfall time series, such as the duration and depth of a rainfall event (Yang and Chui, 2019). However, the information contained within the summary statistics may be insufficient to predict $Y_t$ at sub-hourly time scales.

A rainfall time series may be represented by aggregated rainfall depths recorded during different intervals, which can be of lower dimensions. The rainfall time series recorded between time steps $t-a$ and $t-b$ can be represented by a rainfall depth $D_{t-a,t-b}$ as

$$D_{t-a,t-b} = \sum_{i=a}^{b} P_{t-i}, \tag{8}$$

In this case, $\varphi(\boldsymbol{X_t}) \coloneqq \big[D_{t-a_1,t-b_1}, D_{t-a_2,t-b_2}, \dots, E_1, E_2, \dots, E_k\big]$. However, an approach to optimally define the set of $(a,b)$ pairs to create rainfall depth features is not known *a priori*; thus, various sets of these values must be tested.

This study proposes a simple method to systematically select cut points along the time axis, which form a series of intervals for defining $D_{t-a,t-b}$. As shown in Fig. 1a, the selection of cut points is controlled by three hyperparameters, $m$, $l$, and $n$. $m$: a cut point is placed between time steps $t-m$ and $t-m-1$ such that the rainfall data recorded prior to $t-m$ are considered irrelevant for predicting $Y_t$. $l$: the rainfall data recorded between $t-l$ and $t-0$ are considered to be most relevant for predicting $Y_t$, so a cut point is placed between every two neighboring time steps within this interval. $n$: $n-1$ cut points are

placed between $t - l - 1$ and $t - m$, such that the neighboring cut points correspond to $n$ intervals whose lengths roughly
220 form an arithmetic sequence.

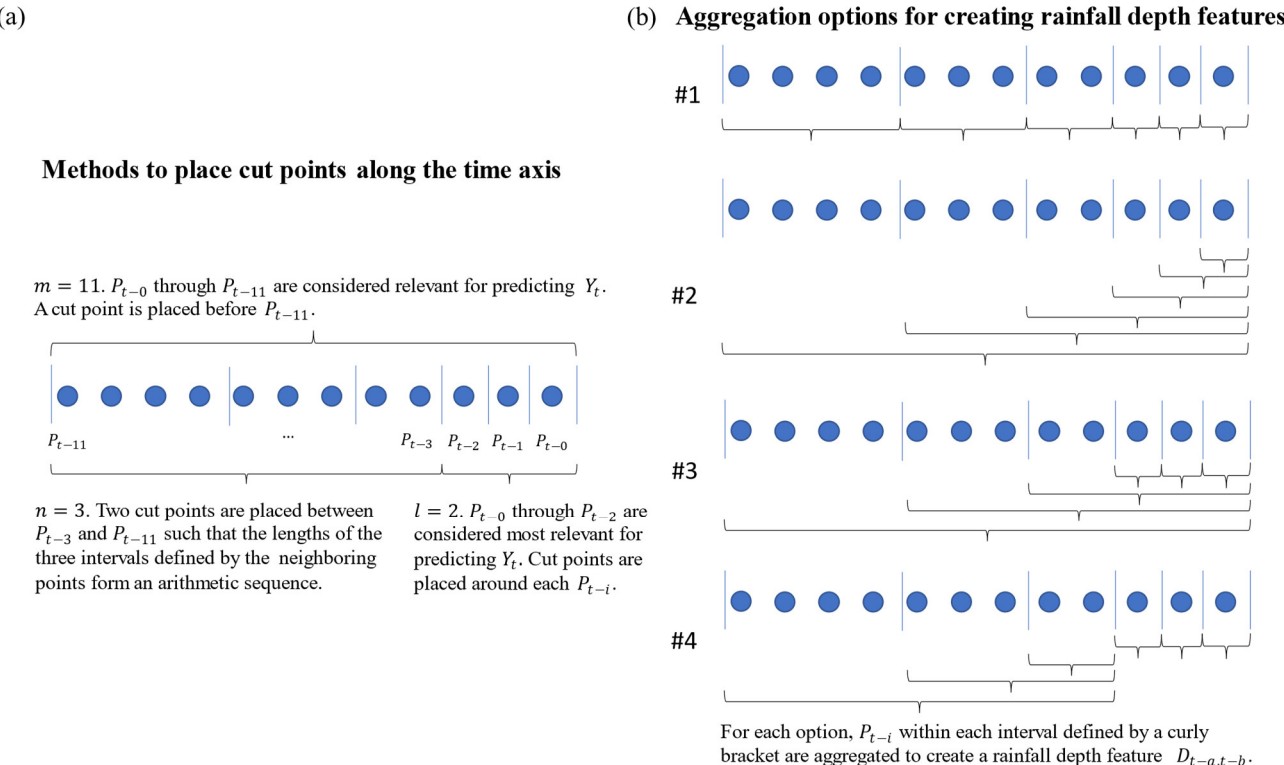

Figure 1 (a) Illustration of the methods to place cut points along the time axis. (b) Illustration of the four aggregation options for creating rainfall depth features after the cut points are selected.

The rainfall depth features can be created in various ways after the cut points are selected. This study considers four
225 aggregation options, as illustrated in Fig. 1b. (1) A rainfall depth feature $D_{t-a,t-b}$ is created for every interval formed by two neighboring cut points. (2) The set of $D_{t-a,t-b}$ created using option (1) is reused with modifications, such that $a$ is set to 0 for all $D_{t-a,t-b}$. (3) The set of $D_{t-a,t-b}$ created using option (1) is modified, such that $a$ is set to 0 for $D_{t-a,t-b}$ with $b > l$. (4) This option is similar to option (3), except that $a$ is set to $l + 1$ for $D_{t-a,t-b}$ with $b > l$.

Representing a rainfall time series using a set of $D_{t-a,t-b}$ can reduce the data dimension at the cost of losing information
230 regarding the temporal rainfall distribution. Note that fewer cut points are selected for rainfalls in the long-term past (e.g., a few days), implying that they play less important roles in predicting $Y_t$. This is reasonable considering the relatively fast response time of SuDS (DeBusk et al., 2011). Gauch et al. (2020) also showed that the hydrometeorological time series recorded in the long-term past can be represented using a coarser temporal resolution in machine learning models built for rainfall–runoff modeling without deteriorating their prediction accuracy. In the proposed method, the three hyperparameters

and aggregation options control the aggregation level and approach by which rainfall data recorded at different time steps are aggregated. The optimal hyperparameters and options are determined using the resampling technique as described below.

### 2.2.3 Contribution of rainfall at each time step to runoff predictions

The SHAP value of a rainfall depth feature $D_{t-a,t-b}$ for a particular input $\mathbf{x_t}$ specifies its contribution to the prediction of $y_t$ in the machine learning model and is denoted as $\emptyset_{D_{t-a,t-b}}(\mathbf{x_t})$. $\emptyset_{D_{t-a,t-b}}(\mathbf{x_t})$ can be distributed among the rainfall data to determine the extent to which rainfalls at each time step contribute to the prediction of $y_t$. Let $d_{t-a,t-b}$ denote the rainfall depth recorded between time steps $t-a$ and $t-b$, and let $p_{t-k}$ denote the rainfall depth of the past time step recorded at $t-k$, where $a \le k \le b$. Because $d_{t-a,t-b}$ is simply the sum of the rainfall depths recorded between $t-a$ and $t-b$ (Eq. (8)), $\emptyset_{D_{t-a,t-b}}(\mathbf{x_t})$ can be assigned to each rainfall depth datum recorded within that period proportional to its value, i.e., $\frac{p_{t-k}}{d_{t-a,t-b}}\emptyset_{D_{t-a,t-b}}(\mathbf{x_t})$ is assigned to $p_{t-k}$ if $d_{t-a,t-b} \ne 0$.

However, $p_{t-k}$ can be used in multiple rainfall depth features in the machine learning model. Its overall contribution to predicting $y_t$ is the sum of the contributions associated with each rainfall depth feature. $\tau_k(\mathbf{x_t})$ is the contribution of the rainfall depth datum recorded at time step $t-k$ and can be computed following

$$\tau_k(\mathbf{x_t}) = \sum_{(a,b)\in A_k} r(k,a,b) * \emptyset_{D_{t-a,t-b}}(\mathbf{x_t}), \tag{9}$$

where

$$A_k = \{(a,b) \in A | a \le k \le b\}, \tag{10}$$

$$r(k,a,b) = \begin{cases} \frac{p_{t-k}}{d_{t-a,t-b}} & \text{if } d_{t-a,t-b} \ne 0 \\ \frac{1}{b-a+1} & \text{if } d_{t-a,t-b} = 0 \end{cases}, \tag{11}$$

where $A$ is the set of $(a,b)$ pairs used to define the rainfall depth features $(D_{t-a,t-b})$ of the machine learning models, $A_k$ is the subset of $A$ with $a \le k \le b$, $p_{t-k}$ is the rainfall depth recorded at $t-k$ for input sample $\mathbf{x_t}$, and $d_{t-a,t-b}$ is the rainfall depth recorded between $t-a$ and $t-b$.

After the contribution of rainfall at each time step is computed, the average rainfall age $\alpha(\mathbf{x_t})$ that contributes to the runoff prediction weighted by importance can be computed as

$$\alpha(\mathbf{x_t}) = \frac{\sum_{k=0}^{m}|\tau_k(\mathbf{x_t})|*k}{\sum_{k=0}^{m}|\tau_k(\mathbf{x_t})|}, \tag{12}$$

The absolute value of $\tau_k(\mathbf{x_t})$ is used here because this term can sometimes be negative, and it is not logical to assign a negative weight to the rainfall age $k$. Thus, $\alpha(\mathbf{x_t})$ should only be interpreted as the average age of the rainfalls that control the runoff prediction, but not the average time that the stormwater spends in the catchment prior to draining, i.e., water age (Walker and Krabbenhoft, 1998).

**2.2.4 Combining local explanations to understand model structures and rainfall–runoff correlations**

Lundberg et al. (2020) showed that the SHAP values of multiple samples can be combined to achieve a global understanding of the model structure and system processes. In this study, the contribution of rainfall at each time step to the runoff prediction, $\tau_0(\mathbf{x_t})$ through $\tau_m(\mathbf{x_t})$, is computed for each sample. The contributions obtained for various collections of samples can be combined to better understand the model structure and rainfall runoff correlation for specific regions in the input domain. For instance, the input samples that correspond to high flow can be collectively analyzed if high flow predictions are of interest, and the global model structure can be inspected by analyzing the explanations for all samples. The average contribution associated with a set of samples can be characterized by its arithmetic mean because contributions are additive (Eq. (4)). The average contribution of $p_{t-k}$ to the runoff prediction among the input samples in set $A$ ($\Gamma_k(A)$) can therefore be computed using

$$\Gamma_k(A) = \frac{\sum_{\mathbf{x}_t \in A} \tau_k(\mathbf{x}_t)}{|A|}, \tag{13}$$

where $|A|$ is the number of elements of $A$.

$\tau_k(\mathbf{x_t})$ can occasionally be negative in some models; positive and negative $\tau_k(\mathbf{x_t})$ values may therefore cancel out when summing the values from multiple samples. The absolute value of $\tau_k(\mathbf{x_t})$ thus indicates the importance of $p_{t-k}$ for the runoff prediction and can be used when aggregating the contributions of multiple samples. The average importance of $p_{t-k}$ for the runoff prediction among samples in set $A$ ($I_k(A)$) can therefore be computed according to

$$I_k(A) = \frac{\sum_{\mathbf{x}_t \in A} |\tau_k(\mathbf{x}_t)|}{|A|}, \tag{14}$$

**2.2.5 Other global interpretation methods**

There exist some commonly used global feature importance measures for different types of machine learning models. Three global feature importance measures that are specifically designed for decision tree-based models are considered in this study to compare with $\Gamma_k$. A typical decision tree has a flowchart-like structure with a root node that connects to a number of leaves (i.e., terminal nodes) via internal-node chains. Each leaf holds a prediction and each non-leaf node specifies a test, where an input sample is $\mathbf{x}$ compared with a threshold value of the feature stored at that node. The test result determines which internal node or leaf $\mathbf{x}$ will be next visited. The prediction made for $\mathbf{x}$ is the value of the leaf it ultimately selects. A detailed introduction to tree-based models can be found in Myles et al. (2004)

The three measures considered in this study are *gain*, *cover*, and *frequency*. The gain of a feature is the relative value of the improvement evaluated at all of the nodes that split on this feature. In the tree-based models used in this study, the improvement of a split reflects the reduction of the objective function that the machine learning algorithm tries to minimize. The gain of a feature can be roughly regarded as its associated improvement of the prediction accuracy (Chen and He, 2020). The cover of a feature is the relative value of the coverage associated with all of the nodes that split on this feature. For the runoff prediction problems examined in this study, the coverage of a node represents the number of samples from the training dataset that reach

this node when passing the entire dataset through the trees. The frequency of a feature is the relative number of times that this feature has been used in the tree nodes to determine the splits. Computation details of these three measures can be found in Chen and Guestrin (2016) and Chen and He (2020).

### 2.2.6 XGBoost algorithm

XGBoost (Chen and He, 2020) is an open-source machine learning software library adopted in this study to train the machine learning models. The resulting models are gradient-boosted trees (Friedman, 2001). XGBoost is selected for its improved regularization methods, high computational efficiency, and ability to achieve state-of-the-art results on many machine learning tasks (Chen and Guestrin, 2016; Chen and He, 2020; Nielsen, 2016). Another reason for using XGBoost is that both the observational and interventional SHAP values of the resulting models can be efficiently computed using the TreeExplainer algorithms proposed by Lundberg et al. (2020) using the model tree structures. The computation of the exact SHAP values can be highly computationally expensive for many other types of machine learning models. The XGBoost algorithm is briefly introduced below and more information can be found in Chen and Guestrin (2016) and Mitchell and Frank (2017).

A gradient boosted trees model $\theta$ built by XGBoost is an ensemble of decision trees, in which $\hat{y}_i$, the prediction for an input $\mathbf{x}_i$, is the sum of the predictions of individual trees (Chen and Guestrin, 2016), given as

$$\hat{y}_i = \theta(\mathbf{x}_i) = \sum_{k=1}^{K} f_k(\mathbf{x}_i), f_k \in \mathcal{F}, \tag{15}$$

where $f_k$ is a decision tree that maps $X$ to the values assigned with the tree leaves, $K$ is the number of decision trees (also known as the number of boosting iterations), and $\mathcal{F}$ is the functional space of all possible regression trees.

The trees are built to minimize the objective function within the following equation

$$\mathcal{L}(\theta) = \sum_{i=1}^{n}(y_i - \hat{y}_i)^2 + \sum_{k=1}^{K} \Omega(f_k), \tag{16}$$

where $y_i$ is the observed outcome of an input $\mathbf{x}_i$, $n$ is the number of samples of $(\mathbf{x}_i, y_i)$ pairs used as the training dataset, and $\Omega$ is a regularization function that penalizes complex trees. There are a number of hyperparameters used in $\Omega$ that specify how much penalization a decision tree should receive. Other hyperparameters, such as $K$ and the maximal tree depth, are also used in XGBoost to control the model structure and learning behavior during training. A complete list of the hyperparameters can be found in the XGBoost software documentation (Chen and He, 2020). The XGBoost hyperparameters and feature engineering hyperparameters are optimized together using a hyperparameter optimization process described below.

### 2.2.7 Nested cross-validation and Bayesian optimization for training and testing machine learning models

Both the feature engineering hyperparameters (described in Sect. 2.2.2) and XGBoost hyperparameters must be set prior to training a machine learning model. In this study, the optimal hyperparameters that minimize the prediction errors are found using a nested cross-validation (CV) procedure with Bayesian optimization, from which the model prediction accuracy is also assessed.

The nested CV is implemented as follows. (1) The dataset of the observed $(\mathbf{x}_t, y_t)$ pairs is split into $v$ datasets of roughly equal size, which are termed folds. (2) During an outer CV iteration, the model with the optimal hyperparameters trained on $v - 1$ folds is tested on the remaining fold. The outer CV iteration proceeds until all of the folds have been used for testing. The procedures to identify the optimal hyperparameters during each outer CV iteration are introduced next. (3) The test errors are used as estimations of the generalization error of the machine learning algorithm, which is the expected prediction error of the unseen data.

During each outer CV iteration, an inner CV procedure is used to identify the optimal hyperparameters, where the dataset of $v - 1$ folds is further split into $w$ folds. The effectiveness of the set of candidate hyperparameters is then assessed by the validation error of the resulting models, which is the mean prediction error associated with each fold when the remaining $w - 1$ folds are used for model training. A function that maps a set of hyperparameters to a validation error can thus be defined. The optimal hyperparameters that minimize the prediction error can be found using various function optimization algorithms.

The function optimization problems are solved using Bayesian optimization methods in this study, which are sample-efficient algorithms for solving black box optimization problems (Shahriari et al., 2016). The sample-efficient property of these methods allows near-optimal solutions to be found with a relatively small number of function evaluations. This is particularly useful for hyperparameter optimization, in which the validation errors can be expensive to compute (Snoek et al., 2012). In this study, the decision variables of the optimization problems are six XGBoost hyperparameters and three to five feature engineering hyperparameters depending on the case studies, as described in the following section. The lower and upper bounds of these variables can be found in the source code provided in the Code availability section. Bayesian optimization methods may be treated as black-box methods that automatically find the optimal solution within the specified search space of the decision variables within the given computational budget. The searching behaviors of the methods are determined by a set of parameters. Their parameter values used in this study are also provided in the source code given in the Code availability section. A detailed introduction to Bayesian optimization can be found in Frazier (2018).

However, it should be noted that hyperparameter optimization processes face the risk of overfitting the model selection criterion (Cawley and Talbot, 2010). During each inner CV iteration, multiple models are trained on the same dataset with different sets of candidate hyperparameters, and their prediction accuracies are evaluated on the same validation dataset. The model with the highest prediction accuracy may generalize poorly to the unseen data because it may simply exploit the randomness of the validation dataset without learning the true correlation between the input and output variables. It is therefore important to test whether the hyperparameters found during the inner CV iterations are still useful for the unseen data. Thus, in this study, the validation error associated with each set of candidate hyperparameters evaluated during the optimization processes is compared against its test error computed for the test dataset of the corresponding outer CV iteration. A positive correlation of the validation and test errors indicates means that the good hyperparameters identified during the inner CV iterations are likely to also be good for the unseen data. More information on the resampling techniques for testing machine learning models can be found in Kuhn and Johnson (2013) and Hastie et al. (2009).

**2.3 Case studies**

Two SuDS sites with different drainage areas, SuDS practice types, and data availabilities are examined in this study. Machine learning models are built for each site to model the rainfall–runoff correlations. The prediction accuracies of these models are compared against linear regression models in addition to a process-based model for one site. The basis of the predictions is analyzed using the interpretation methods described above, and the obtained explanations for the two sites are compared to validate the model structures and better understand the rainfall–runoff correlations.

**2.3.1 Study sites**

Study site 1 is located on Washington Street, Geauga County, Ohio, U.S., hereafter referred to as "WS." Multiple types of SuDS were built in WS to treat stormwater runoff generated by a nearby commercial building and parking lot, as shown in Fig. 2a (Darner et al., 2015). Runoff from approximately half of the commercial building roof (i.e., an impervious area of 316 $m^2$) drains into a rain garden with a surface area of 37 $m^2$. The 762 $m^2$ parking lot was constructed using porous pavements to allow infiltration.

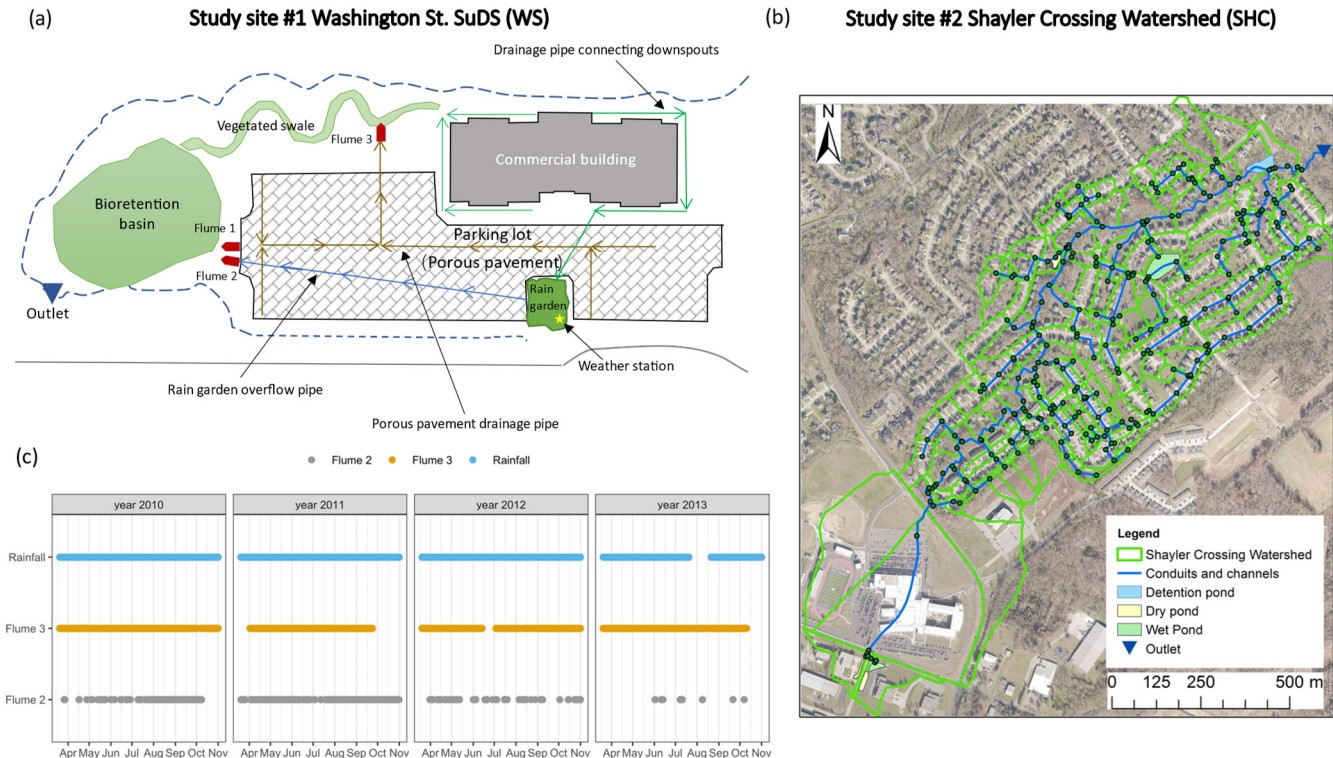

**Figure 2 (a) Layout of the SuDS and monitoring network on the Washington Street site (WS), Geauga County, Ohio, U.S. This figure is adapted from Darner et al. (2015). (b) Map of the Shayler Crossing Watershed (SHC). The subcatchment boundaries and drainage system shown on the map are defined by Lee et al. (2018a). (c) Availability of rainfall and flow rate records collected at regular 10-minute intervals.**

Study site 2 is located in the Shayler Crossing Watershed (SHC) in Clermont County, Ohio, U.S., as shown in Fig. 2b. The SHC is a sub-watershed of the East Fork Little Miami River Watershed. The SHC drainage area is approximately 0.92 km$^2$ (Hoghooghi et al., 2018) and the land use type is primarily residential. The SHC drainage system consists of conduits, channels, detention ponds, dry ponds, and wet ponds (Lee et al., 2018a). In the SHC, stormwater runoff indirectly generated by connected impervious areas (e.g., sidewalks) is treated by the nearby pervious areas, which are termed buffering pervious areas and have similar functions to grass filter strips (Lee et al., 2018b).

Rainfall and discharge time series data are available for both sites. In WS, a 10-min-resolution rainfall time series is available from on-site monitoring between 2009 and 2013. The outflow from WS was collected by three flumes, inside which the water levels were measured every 1 min and recorded every 10 min and when the water level changed. Flumes 1, 2, and 3 respectively collect the surface runoffs from the parking lot, overflow from the surface layer of the rain garden, and underdrain flows from the parking lot. The water levels are converted into discharge measurements using stage-discharge rating curves. Recorded flow data from 2009–2013 are available. The onsite monitoring was conducted by the United States Geological Survey, as described in more detail in Darner and Dumouchelle (2011) and Darner et al. (2015). In the SHC, 10-min-resolution rainfall time series from 2009–2010 are available, in addition to a 10-min-resolution discharge time series measured at the outlet between July and August 2009. The dataset used in this study is the same as in Lee et al. (2018a) and Lee et al. (2018b), in which more details on the dataset can be found.

Process-based models can be difficult to develop for both sites. In WS, the physical properties and exact design of the different drainage system elements are not precisely known (Darner et al., 2015). The rain garden is also not isolated from the gravel storage layer of the porous pavements, which permits an unknown amount of stormwater from the rain garden into the underdrain system of the porous pavement. However, commonly used process-based models are mostly designed to model SuDS with standard designs and may not be directly applicable to WS. In the SHC, the main challenge lies in the heavy workload and uncertainties in estimating the model parameters that characterize the complex drainage system. The SHC can be divided into multiple subcatchments connected by the drainage network, and a number of parameters must be determined for each catchment. In particular, the portions of impervious area that are directly and indirectly connected to the drainage network must be specified to accurately represent the flow paths of each subcatchment. The pervious areas should also be subdivided into areas that treat or do not treat runoff (Lee et al., 2018a). The task of the sub-area division requires substantial effort, considering the relatively large number of subcatchments involved. The numerous parameters associated with each sub-area (e.g., surface roughness, depression storage) can only be determined via model calibration, which is subject to considerable uncertainties.

The two study sites differ in several aspects, including size, SuDS practice type, drainage systems configurations, and data availability. Both sites also face challenges in the application of process-based models and are thus chosen as typical examples of the various SuDS projects to demonstrate the wide applicability of the proposed methods. The configurations of the two study sites are compared in Table 1. More example applications of other SuDS sites can be found in the link provided in the Code availability section.

**Table 1 Configurations of the two study sites.**

|  | Study site #1 | Study site #2 |
|---|---|---|
| Name | Washington Street SuDS site | Shayler Crossing Watershed |
| Abbreviation | WS | SHC |
| Location | Geauga County, Ohio, U.S. | Clermont County, Ohio, U.S. |
| Catchment area | 0.0011 km$^2$ | 0.92 km$^2$ |
| Land use type | Mainly commercial | Mainly residential |
| SuDS implemented | Green roofs and porous pavement | Buffering pervious areas that function similarly to grass filter strips |
| Drainage system configurations | A rain garden was built to treat runoff generated by a commercial building. A porous-pavement parking lot was built near the commercial building. Runoff from the rain garden and porous pavements are collected by a nearby bioretention basin and vegetated swale (not a part of WS). | A drainage network was built to collect stormwater runoff generated in different areas of the watershed. The drainage network consists of conduits, channels, detention ponds, dry ponds, and wet ponds. The indirectly connected impervious areas near the residential buildings are first treated by buffering the pervious areas. |
| Data availability | Flow data collected at the three flumes between 2009 and 2013 are available. The flow rates are derived from water level measurements using the stage-discharge rating curves. The water level at each flume was measured every minute and recorded at 10-min intervals and when the water level changed. A 10-min-resolution rainfall time series recorded between 2009 and 2013 is available. | Flow data collected at the SHC outlet at 10-min intervals from July to August 2009 are available. A 10-min-resolution rainfall time series recorded between 2009 and 2010 is available. |
| Challenges to developing process-based models | The physical properties of the catchment and SuDS are not precisely known. Unknown leakage from the rain garden to the storage layer of the porous pavement is difficult to represent in commonly used process-based models. | Many parameters are required in process-based models to characterize the drainage processes of the complex drainage system. A considerable amount of data on the physical properties of the catchment is required. Some parameters can only be determined through model calibration, which is subject to large uncertainties. |
| Hydrological models built in previous studies | The authors are not aware of any process-based model built for this site. | Lee et al. (2018a) built a process-based model for this site using SWMM software (Rossman, 2015). |

## 2.3.2 Numerical experiments

Rainfall–runoff models are built for WS and SHC. In WS, the output variable is the flow rate of the total runoff collected by the three flumes recorded at regular 10-min intervals. The input variables include the 10-min-resolution rainfall time series recorded prior to runoff, the month in which the runoff occurs (optional), and the accumulative rainfall depth recorded since the beginning of monitoring (optional). The optional features are considered to account for the possible evolving performance of the SuDS during its service life (Yong et al., 2013) and potential seasonality of the SuDS hydrological properties (Muthanna et al., 2008). Whether the two sets of optional features should be included is controlled by the two binary feature engineering hyperparameters, in addition to $m$, $l$, and $n$ described in Sect. 2.2.2. The rainfall–runoff data recorded during 2010–2013 for the warm season (i.e., April to October) (Darner et al., 2015) are used for modeling in this study. Fig. 2c shows the temporal distributions of the available data. Flows were not observed at flume 1 throughout the monitoring period and are therefore not shown. The gaps in the runoff data in Fig. 2c correspond to missing values owing to technical issues or extended dry periods with no records. However, the exact cause of each missing value is unknown. Because overflow from the rain garden occurred infrequently (Darner et al., 2015), the missing flow rates of flume 2 are assumed to be 0. The total discharge is then computed for cases in which the flow rates of flume 3 are available. A large gap in the rainfall data can be observed in the summer of 2013. The rainfall time series recorded before the large gap is used for modeling in this study, and the few missing values are assigned to 0.

The feature engineering and XGBoost hyperparameters are automatically optimized using the Bayesian optimization methods and the performance of the resulting models is evaluated using a nested CV procedure, as described in Sect. 2.2.7. For WS, a 5-fold CV is used for both the inner and outer CV iterations. A total of 142 independent rainfall events are identified, defined as rainfall events separated by at least a 24-h dry spell (Guo and Senior, 2006), with 38,835 discharge records during the wet periods (i.e., periods of rainfall lasting at least 24 h), in which the rainfall time series is available for at least the previous 10 days. CV folds are created for the 38,835 rainfall–runoff records using the rainfall event-grouped stratified sampling method (Zeng and Martinez, 2000). The data associated with the same rainfall event are grouped and contained only within a single fold during each of the outer or inner CV iterations, and the peak discharge associated with the rainfall events in each fold roughly follows the same distribution. This is to prevent data leakage and ensure that the data in each fold are representative (Kuhn and Johnson, 2013).

In the SHC, the output variable is the watershed outlet discharge measured at 10-min intervals, and the input variable is the rainfall time series proceeding the discharge measurement. Only two months of runoff data are available with 8,921 discharge records. There are eight missing discharge records during the two-month period, which are excluded from the modeling. The nested CV procedure is not used due to the small dataset size; instead, the dataset is split into training, validation, and test datasets that each contain at least one large runoff event. The Bayesian optimization methods are then used to identify the optimal feature engineering and XGBoost hyperparameters that minimize prediction error on the validation dataset when the

model is trained on the training dataset. The training and validation datasets are then combined and the machine learning methods with the optimal hyperparameters are applied. The resulting models are then tested on the test dataset.

The prediction accuracy of the XGBoost models is compared against linear regression models. For each site, the resampling and hyperparameter optimization methods for training the XGBoost models are also applied to derive linear regression models. The only difference in the training processes between the two model types is that only the feature engineering hyperparameters are used when fitting linear regression models to the data.

For the SHC, the process-based model developed by Lee et al. (2018a) is also compared with the machine learning models built in this study. Their model was built using SWMM software, in which the SHC was divided into 191 subcatchments and the drainage processes in each subcatchment were characterized by further division into sub-areas, such as indirectly connected impervious areas and pervious buffering areas. Lee et al. (2018a) used a considerable amount of data concerning the SHC physical properties to develop this spatially refined model.

After testing the model prediction accuracies, the basis of each runoff prediction is analyzed using the methods described above (i.e., derivation of the rainfall contribution at each time step to the runoff prediction). The explanations of many samples are then combined in different ways to understand the model structure and modeled processes. (1) The explanations of all of the samples are aggregated using Eq. (14) to derive the global feature importance. The feature importance measures are then compared with the gain, cover, and frequency. The feature importance of WS and SHC is also compared to examine whether the feature importance is associated with the modeled hydrological processes. (2) The explanations of the samples are grouped and aggregated based on the predicted discharge to investigate whether the prediction of low and high flows are determined by the same factors, and whether the model structures that lead to these predictions are logical.

The basis of the individual runoff predictions is analyzed at the runoff event levels. The rainfall contributions at different time steps are available for each runoff prediction. This information can then be used to decompose the hydrograph into sub-hydrographs associated with the rainfall at different time steps. This resolves some of the hydrograph separation tasks (Pelletier and Andréassian, 2020), such as identifying the runoff volumes associated with a rainfall event and determining the portions of flow contributed by "old water" (i.e., water stored in the catchment prior to the rainfall event) (Cartwright and Morgenstern, 2018). The average age of the rainfall contributing to runoff prediction at each time step is also computed using Eq. (12).

# 3 Results and discussion

## 3.1 Prediction accuracy of the machine learning models

The Bayesian optimization methods are implemented to automatically identify the optimal feature engineering and XGBoost hyperparameters that minimize the prediction errors of the validation datasets. It is necessary to check whether the prediction accuracies achieved for the validation datasets can be generalized to the unseen data when applying the optimal hyperparameters (i.e., whether or not the model selection criterion is overfitted) (Cawley and Talbot, 2010). Fig. 3 shows the prediction errors associated with each set of candidate hyperparameters obtained during the inner and outer CV iterations in

WS (i.e., validation and test errors). The prediction errors are measured by the root-mean-square error (RMSE). Each subplot shows the results obtained during an outer CV iteration when a specific aggregation option was applied for creating the rainfall depth features. The inner and outer CV errors are positively correlated for all of the outer CV iterations and aggregation options, which is desirable. This result confirms that the hyperparameter optimization process indeed found hyperparameters that perform well on unseen data and the model selection criterion is not overfitted. Similar positive correlations between the validation and test errors are observed in the SHC when different aggregation options are adopted for creating the rainfall depth features.

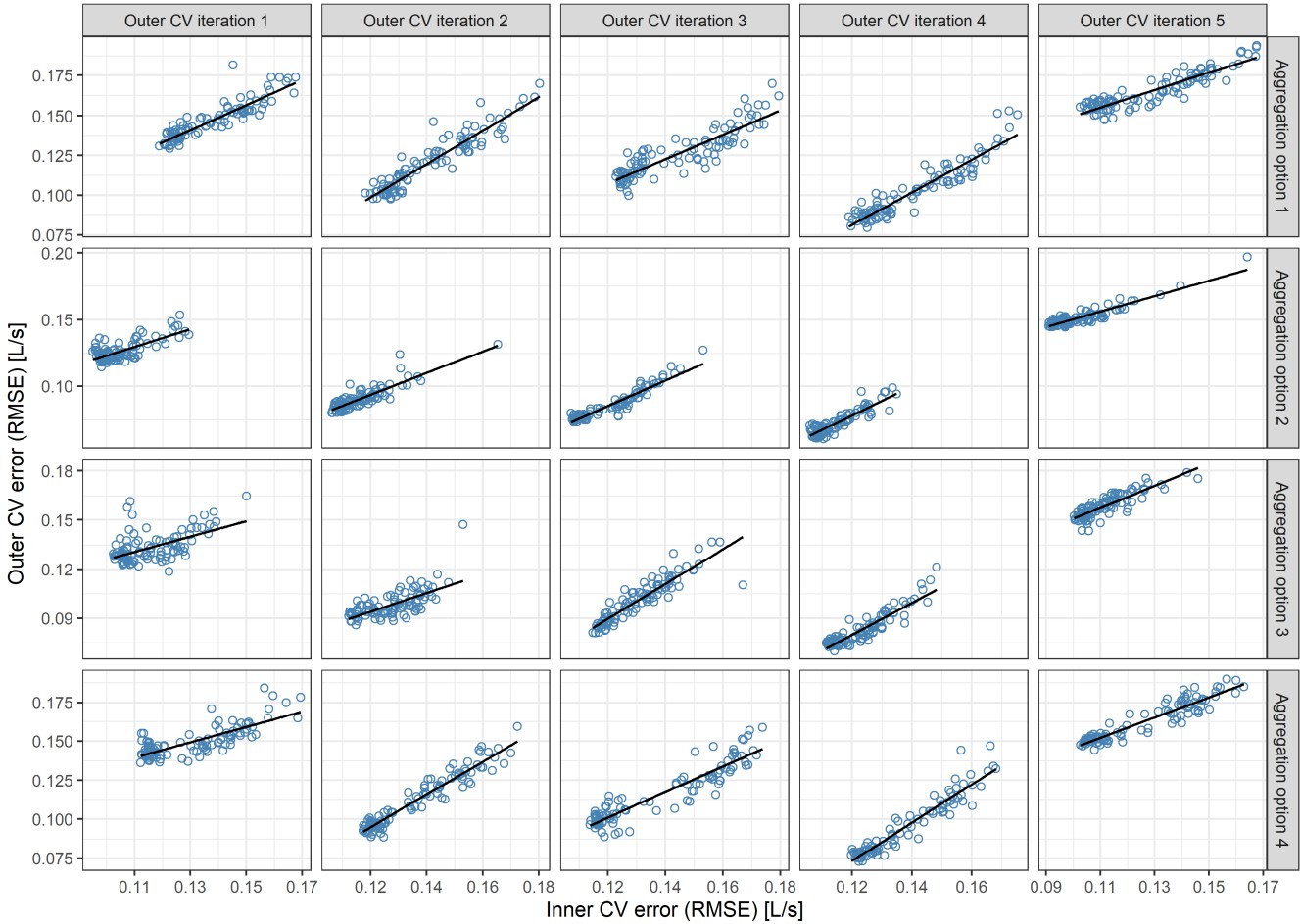

**Figure 3 Prediction errors of the XGBoost models estimated during the inner and outer cross-validation (CV) iterations for each set of candidate hyperparameters evaluated during the optimization processes in the Washington Street SuDS site (WS). The prediction errors are measured by the root-mean-square error (RMSE). Each subfigure shows the result obtained for a rainfall depth feature aggregation option and during an outer CV iteration, and each dot corresponds to a model.**

The prediction accuracies of the various WS and SHC models are compared in Fig. 4. The RMSE, Nash–Sutcliffe model efficiency coefficient (NSE; Nash and Sutcliffe, 1970), and coefficient of determination ($R^2$) of the predictions on the test datasets are compared, except for the SWMM model developed by Lee et al. (2018a), which was tested on a part of its training

dataset due to insufficient data. The prediction accuracies of the XGBoost models, i.e., NSE > 0.7 and $R^2 > 0.7$, can be considered satisfactory, considering that they were relatively easy to set up and that it was impossible or very difficult to build process-based models for either site. The XGBoost models for both sites consistently outperform the linear regression models (LM), suggesting that more sophisticated machine learning algorithms (e.g., XGBoost) are able to better capture complex rainfall–runoff correlations than simple linear regression methods. The SHC XGBoost models have comparable prediction accuracies to SWMM, although the former were built with considerably less data.

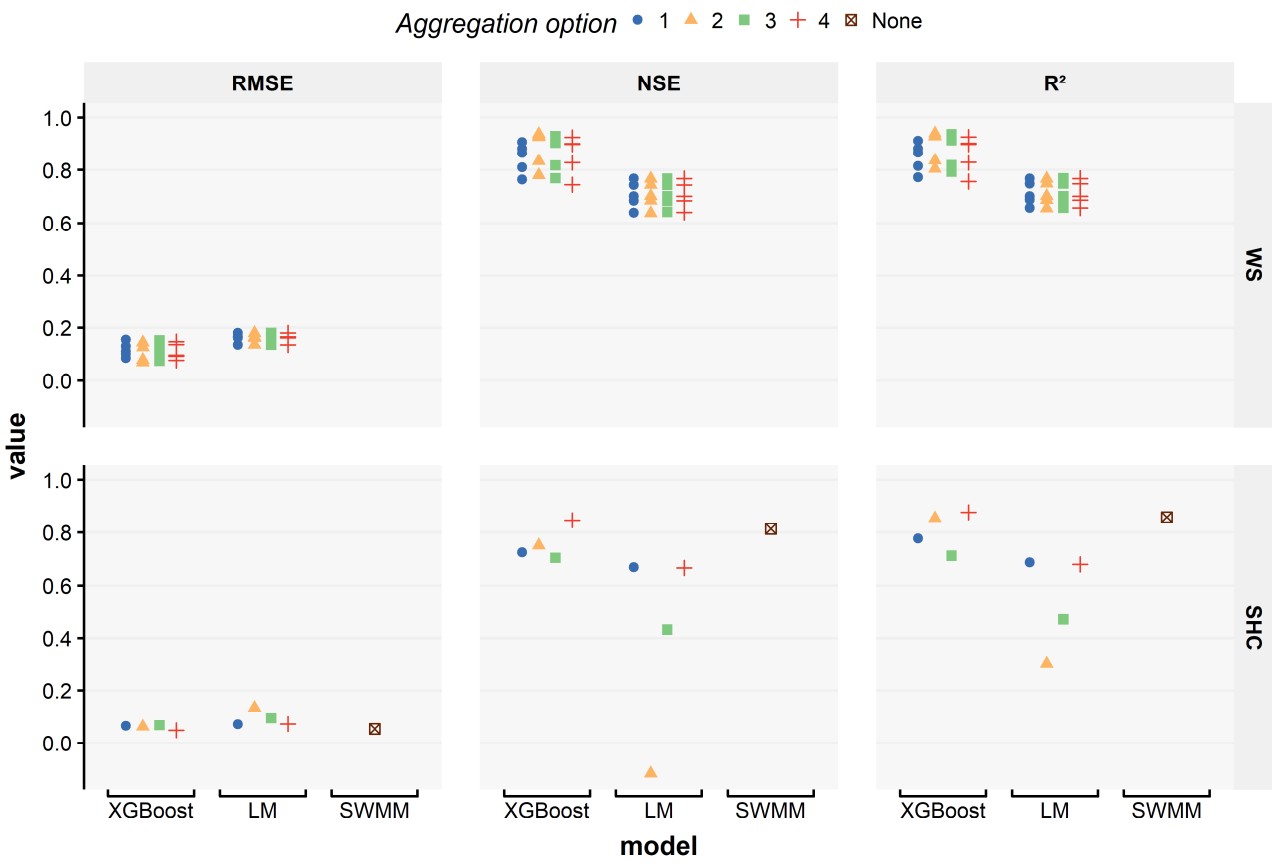

**Figure 4 Prediction accuracies of the various models built for the Washington Street SuDS site (WS) and Shayler Crossing Watershed (SHC). The prediction accuracies are evaluated in terms of the root-mean-square error (RMSE), coefficient of determination ($R^2$), and Nash–Sutcliffe coefficient of efficiency (NSE). The RMSE units are L/s for WS and m³/s for the SHC. Each data point in the figure shows the prediction accuracy of a model tested on the associated test dataset. All of the SHC models were tested on the same dataset, and nested cross-validation was not implemented due to the small sample size. The SWMM model for SHC was built by Lee et al. (2018a).**

The four aggregation options for creating rainfall depth features generate models with similar prediction accuracies for both sites. Thus, all of the aggregation options may be useful. In practice, an ensemble of models produced using different feature engineering methods may be used.

Each data point in Fig. 4 shows the result obtained for a machine learning model that was trained and tested on a pair of training and test datasets. In WS, considerable variations in the prediction accuracies can be observed when a machine learning method is applied to different training and test datasets owing to the different resulting model structures and model performance assessment criteria resulted from different test datasets. The variations indicate that the sample distributions in the different versions of training and test datasets appreciably differ, even though a stratified sampling method is used to balance the sample

distribution in the different folds. The imbalanced sample distribution is associated with the limited number of samples used for the model training and evaluation, which implies that the four years of rainfall–runoff data still contained an insufficient number of samples for the different regions of the input space in the machine learning methods examined in this study. For instance, only a few high-flow points were observed each year, which may be insufficient for the training machine learning models to provide accurate high-flow predictions. Even fewer samples were available for training and testing the SHC machine

learning models; thus, the uncertainties of the prediction accuracies may be even larger.

     Given that the uncertainties in the prediction accuracies are not negligible, the model assessment results should be interpreted carefully. It may be useful to examine whether the model structures that lead to each prediction are logical, which is examined in the following sections.

## 3.2 Understanding the global model structures for rainfall–runoff modeling

The predicted discharge for a specific input sample can be decomposed into the rainfall contributions of different time steps using Eq. (9) to Eq. (11) based on the SHAP values assigned to the rainfall depth features. The absolute value of these contributions specifies the importance of the rainfall for the runoff prediction. The contributions and importance associated with all of the samples can be averaged to understand the global model structures for making discharge predictions using Eq. (13) and Eq. (14). Fig. 5a and 5b respectively show the average importance and contributions of the rainfall at each time step

in the past for predicting the discharge at the current time step in the different models of WS. Each subplot displays the results obtained using a specific aggregation option for creating the rainfall depth features and a specific definition of the SHAP values (i.e., Eq. (2) or Eq. (3)). Fig. 5a shows that in all models, the discharge prediction is mostly affected by more recent rainfalls (note that the x-axis is on a pseudo-logarithmic scale and each time step is 10 min). The rainfalls recorded prior to 100 time steps in the past (i.e., 16.7 h) have almost no impact on the discharge prediction. This result is reasonable considering the small

WS catchment size. However, Fig. 5b shows that on average, the more recent rainfalls have a negative contribution to the discharge prediction. The negative values indicate that the more recent rainfalls can have a negative contribution to the runoff predictions across the samples in the machine learning models. This contradicts the conventional process-based models, where rainfall positively affects the runoff generation processes.

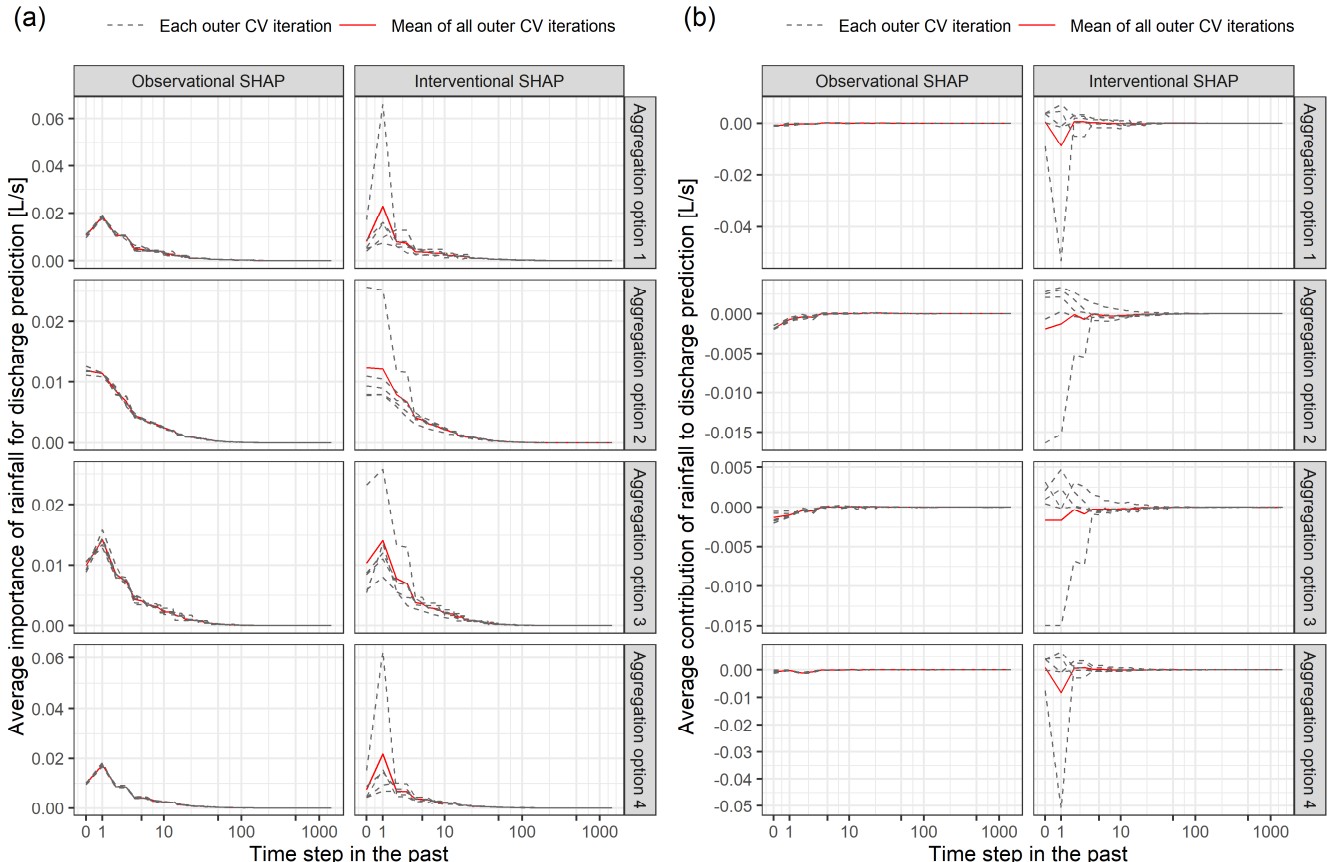

Figure 5 Average (a) importance and (b) contribution of the rainfall at different time steps in the past for the discharge predictions of different Washington Street SuDS site (WS) models. Each subfigure shows the results obtained using a specific aggregation option for creating the rainfall depth features and a SHAP definition. Each line in the figure corresponds to the results obtained for a machine learning model trained during an outer cross-validation iteration or the average values computed for all of the models.

Each subfigure of Fig. 5 shows the results obtained for different versions of the models derived in the outer CV iterations of each aggregation option. Across different versions of the models, the rainfall at each time step is generally found to have similar a contribution and importance to the discharge prediction, which indicates a high degree of similarity between the rainfall–runoff correlations learned by these models. However, the rainfall can be assigned notably different importance and contributions values when different aggregation options are adopted, which implies that the rainfall–runoff correlations are modeled differently in these models. Schmidt et al. (2020) referred to the existence of various possible model structures of machine learning models as equifinality, which is an important concept in hydrological modeling (Beven and Freer, 2001).

Both the observational and interventional SHAP values are used to compute the rainfall contributions and importance values. As mentioned in Sect. 2.1.2, the resulting values should be interpreted differently. The contributions resulting from the observational SHAP values can be interpreted as the expected difference in the predicted discharge when a particular rainfall is observed/not observed, accounting for the presence/absence of the other rainfall measurements. The importance is the

absolute value of the contributions, which measures the extent of the effect. The contributions that result from the interventional SHAP values are the expected prediction changes when rainfall is set to a specific value, accounting for the presence/absence of other rainfall measurements. The importance is also the absolute value of the contribution. As shown in Fig. 5, the contributions and importance values resulting from the interventional SHAP values have larger variabilities among the different outer CV iterations, which suggests that the interventional SHAP values are more sensitive to the small model

structure variations. Nevertheless, the observational and interventional SHAP values produce similar results in terms of the relative importance of the rainfall at different time steps for the runoff prediction.

The average rainfall contribution and importance for the discharge prediction are also computed for SHC, as shown in Fig. 6. The runoff predictions are mostly affected by the rainfalls observed in the periods approximately nine time steps in the past (i.e., 1.5 h). In contrast, the discharge prediction of the SHC is mostly affected by the rainfall from the past 1 time step (i.e.,

10 min). Because the SHC is considerably larger than WS, it is expected to have a longer concentration time (i.e., longer required time for stormwater to travel through the catchment). It is therefore reasonable that the models show that the discharge predictions in the SHC are mostly affected by rainfall from the more distant past than those in WS. It is also worth noting that the contribution and importance assigned to the rainfall do not vary smoothly across the time steps, which indicates that the models find rainfall in some periods to be more important than other periods, which is not realistic. These undesirable model

behaviors, which were not observed in the WS models, might be caused by the insufficient data used in the model training.

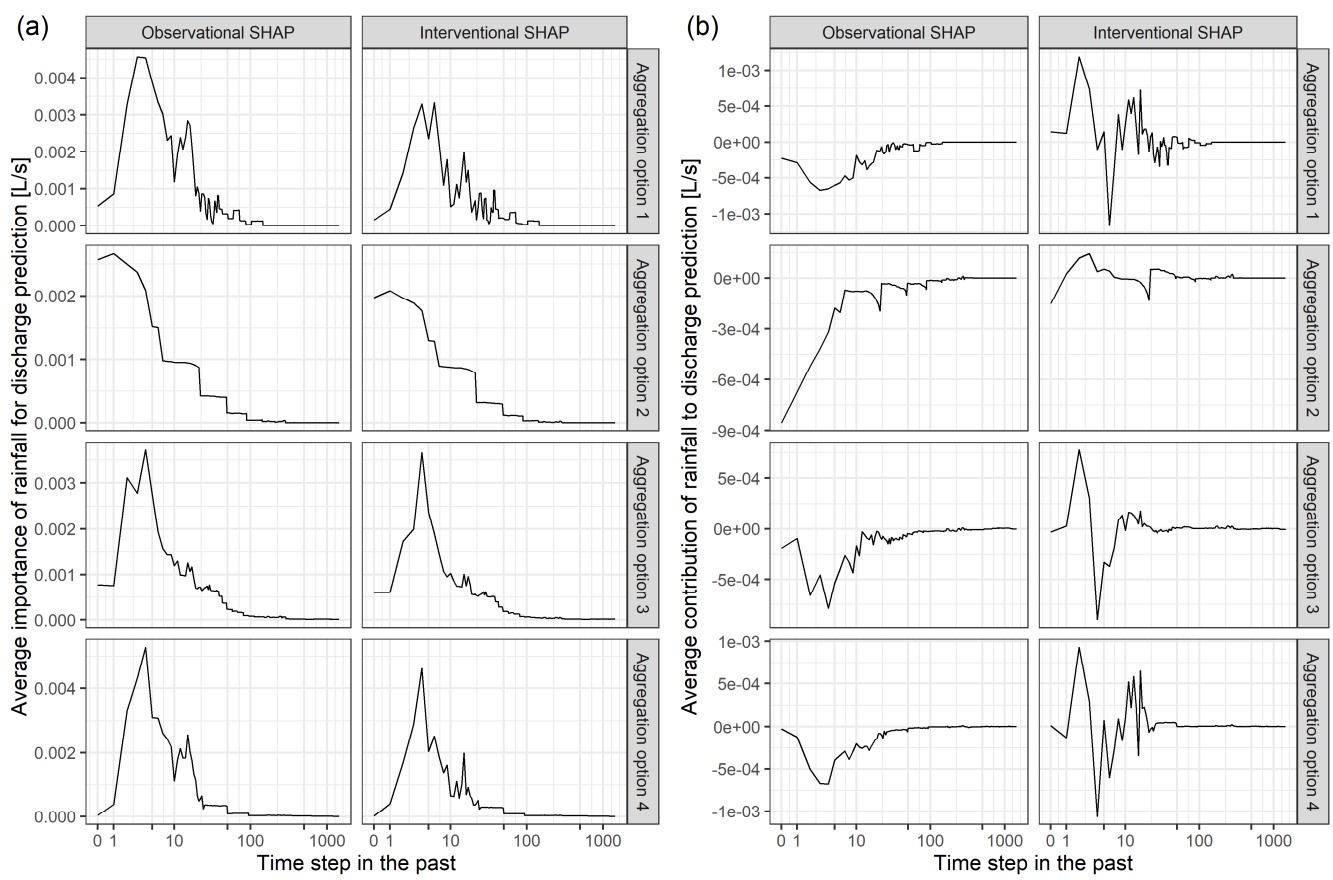

**Figure 6 Average (a) importance and (b) contribution of the rainfall of different time steps in the past for the discharge predictions in different models of the Shayler Crossing Watershed (SHC). Each subfigure shows the results obtained using a specific aggregation option for creating the rainfall depth features and a SHAP definition.**

Two sets of optional features (i.e., seasonality and accumulative rainfall depth) are considered in WS; however, these features are infrequently selected in the hyperparameter optimization processes (both were selected 5 times in 20 models of different aggregation options and outer CV iterations), and their combinations were found to affect the obtained prediction by up to 12.4% and 10.3% according to the average observational and interventional SHAP values of all features. This suggests that optional features are less important for discharge predictions.

The gain, cover, and frequency metrics are computed for each input feature of each model. Each of the three feature importance measures of each rainfall depth feature ($D_{t-a,t-b}$) are then distributed among the rainfalls of the different time steps ($P_{t-i}$) that are used to define $D_{t-a,t-b}$ (as specified in Eq. (8)) proportional to their values in all of the input samples. For instance, the gain of $D_{t-0,t-1}$ is distributed among $P_{t-0}$ and $P_{t-1}$ proportional to their values in all samples. The average importance of rainfall of each time step for discharge prediction is then computed using the results of all samples for the three

feature importance measures. The SHAP value-based feature importance as defined in Eq. (14) is also computed for rainfall of each time step. A normalization procedure is applied in which the importance of the rainfall of each time step is divided by

the sum of the average importance of all features (including both the rainfall depth features and the other features). Fig. 7 compares the various feature importance measures of rainfall of each time step of the WS models. In general, the feature importance is more unevenly distributed across time steps when gain metrics are used than the cover and the frequency metrics.

This result suggests that although the frequency of the rainfall of each time step used by the models to make predictions is similar, the more recent rainfalls are associated with greater contributions to improving the prediction accuracy. Results of the two SHAP value-based feature importance measures are generally similar to that of the other commonly used importance measures. These similarities confirm the validity of the SHAP values in identifying the important features. In addition, feature importance measures computed using various methods are also found to be relatively similar in the SHC.

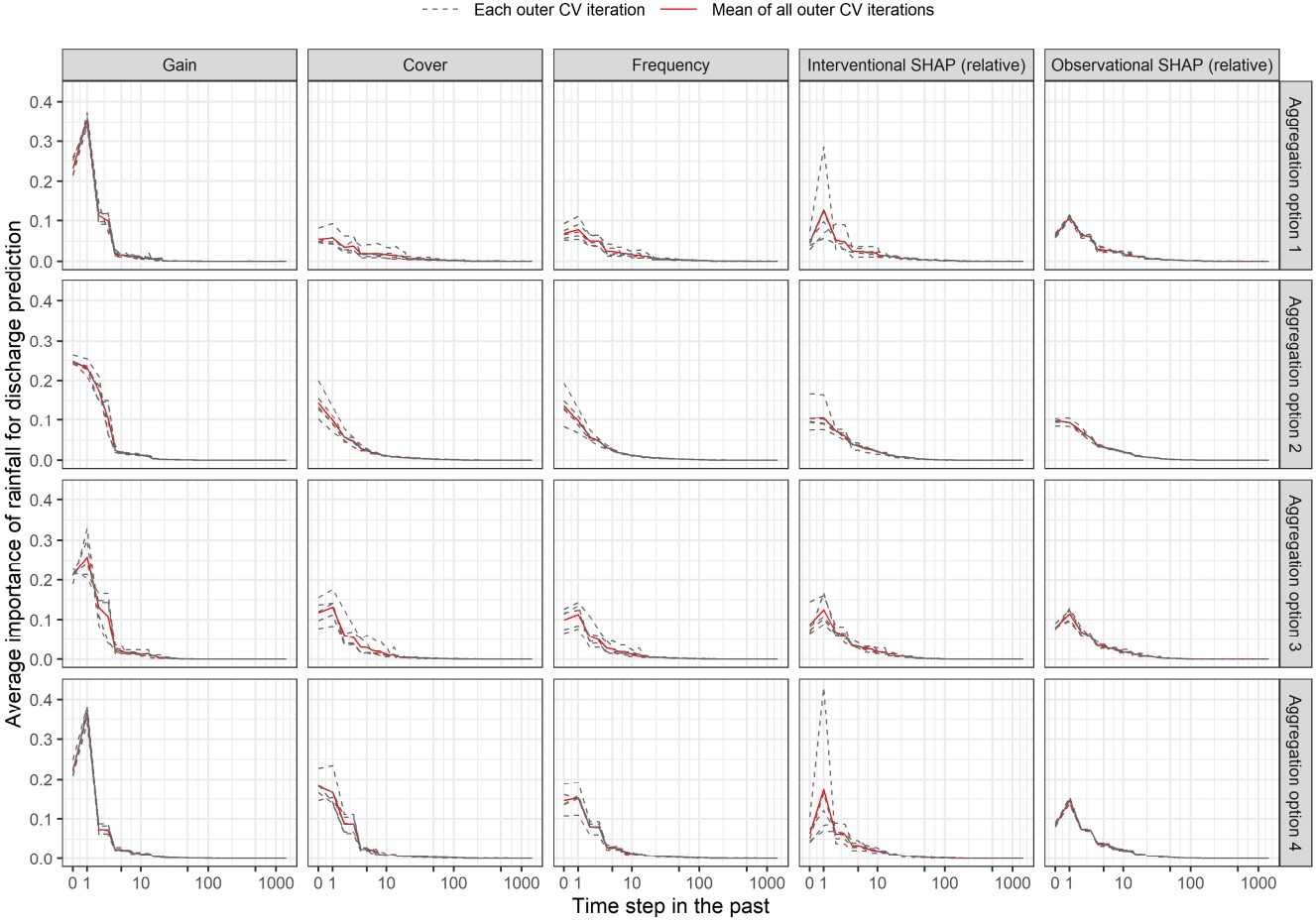


**Figure 7 The importance of rainfall from each time step for making discharge predictions assessed by different feature importance measures in the Washington Street SuDS site (WS) models. Each subplot shows the results obtained for the models built with an aggregation option to create rainfall depth features and a feature importance measure. Each line shows the results of model trained during an outer cross-validation iteration or the average results obtained in all outer CV iterations.**

## 3.3 Understanding the model structures for predicting discharges of different magnitudes

Unlike the commonly used global feature importance measures, the SHAP values are local feature attribution methods, in which the basis of each prediction made by a model can be explained. The local explanations can be combined to understand the models' behavior in specific regions of the input domain. As an example, this study examines whether the low- and high-flow predictions of the models are controlled by similar factors. The input variables are clustered into five groups according to the magnitude of the predicted discharge. Each group contains roughly the same number of samples. Samples in the 0–20th percentile interval form the first group, samples in the 20–40th percentile interval form the second group, and so on. The average importance and contribution of the rainfall of each time step are computed for the samples in each group. The results obtained using rainfall depth feature aggregation option 1 and the observational SHAP values are shown in Fig. 8. The results show that the discharge predictions are mostly controlled by recent rainfalls, regardless of the magnitude of the predicted discharge.

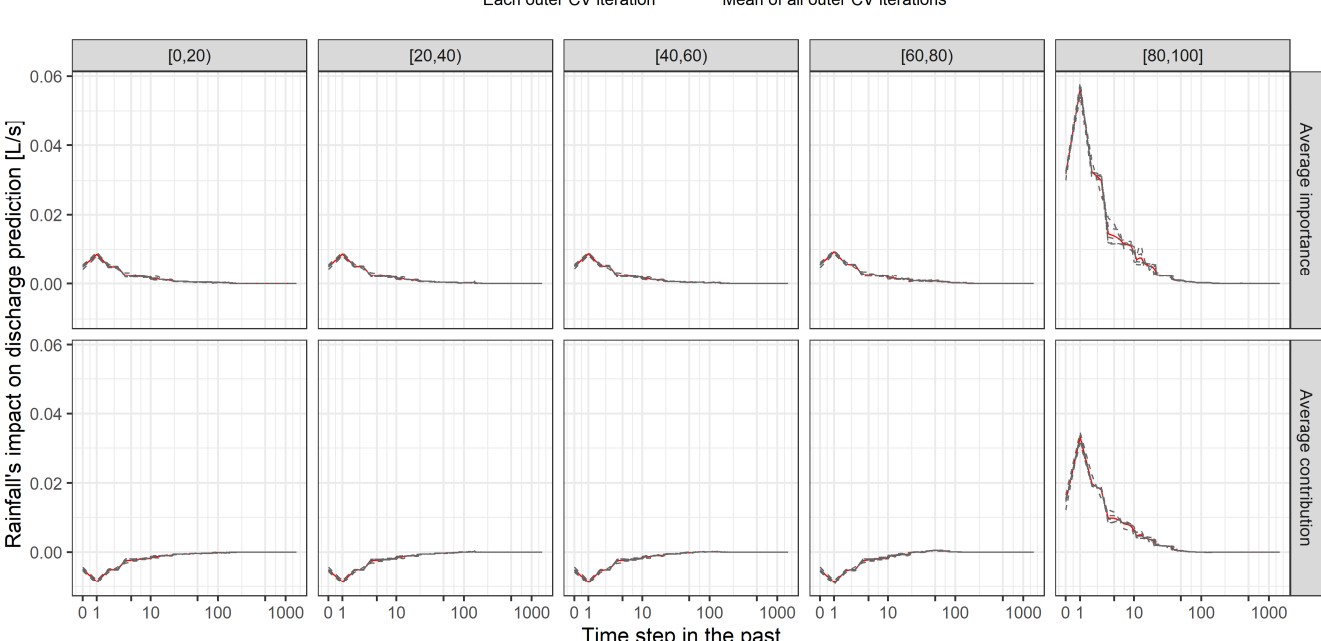

**Figure 8 The average importance and average contribution of rainfall from each time step in the past for predicting discharges of various magnitude in the Washington Street SuDS site (WS) models using the aggregation option. Each subfigure shows the results for the discharge prediction in different percentile ranges. The subplot labels show the percentile range. Each line shows the results of the model trained during an outer cross-validation iteration or the average results obtained in all of the outer CV iterations.**

It is interesting to note that the recent rainfalls in Fig. 8b have a negative contribution to the discharge prediction in small to medium discharges (i.e., 0–80th percentiles). This means that the discharge predictions would be higher if information regarding the recent rainfall depths were unavailable. The implication is that the models have certain threshold values, such that the recent rainfalls have non-negative contributions to the discharge predictions only if they are larger than a given

threshold. This also implies that discharge may be overpredicted for low to medium flows when based solely on rainfall information from the relatively distant past. These properties correspond to a conceptual rainfall–runoff model that differs from other commonly used models in hydrology, such as the explicit soil moisture accounting models (Beven, 2012), where rainfall cannot have a negative contribution to runoff. In machine learning models, however, the catchment is associated with a basic discharge prediction that can be negative, and the rainfalls are compared to some threshold and will only have a positive contribution if they are larger than the threshold and vice versa. This representation can be roughly expressed as Eq. (17), which has a similar form to linear regression models.

Discharge prediction = basic prediction + (rainfall depths − thresholds) ∗ transformation factors,           (17)

This model can be referred to as a company acquisition model, in which the company (i.e., basic prediction) is in the business of acquiring companies of various values (i.e., rainfall at different time steps), which can be negative. The total value (i.e., discharge prediction) of the merged company is determined by the size of each company and how each company interacts.

Negative rainfall contributions were also found for the low-flow predictions in some of the models built using other rainfall depth feature aggregation options for WS and the SHC.

Fig. 8 shows that the weights assigned to the rainfall of each time step change smoothly across time steps to predict runoffs of various magnitudes. This indicates that the rainfall–runoff relationship learned by the models is stable for different regions of the input domain.

### 3.4 Hydrograph decomposition according to the contribution of rainfall of each time step to runoff prediction

The contribution of runoff at each time step to the discharge prediction can be useful for hydrograph decomposition, as each discharge prediction equals the sum of the SHAP values of all of the input sample features in addition to some constant bias (i.e., local accuracy properties described in Eq. (4)). As an illustration, Fig. 9 shows the decomposed hydrographs for large, medium, and small runoff events predicted by a machine learning model of WS. The rainfalls collected at 10-min intervals are aggregated to rainfall depths recorded between the past 0–1 h, 1–2 h, and so on. The contribution associated with each rainfall depth record is also accordingly aggregated. Fig. 9 shows that runoff is mostly affected by the rainfall that occurred in the past 1 h regardless of the runoff event magnitude, especially for the peak discharge. However, this hydrograph decomposition method has some limitations. First, negative contributions are assigned to the rainfalls, which are difficult to interpret. Second, there is a constant bias term in the hydrograph, which does not correspond to any element in the commonly used conceptual rainfall–runoff models. Third, a model might use features that are not derived from rainfall (e.g., temperature) as a predictor, which will also be assigned with contributions to the runoff predictions when SHAP methods are used. It is unclear how to represent the contributions of factors other than rainfall when decomposing a hydrograph. Nevertheless, the hydrograph decomposition results shown in Fig. 9 are still useful for understanding why a given prediction is made by the model and to what extent the rainfall of each time step contributes to the model runoff prediction.

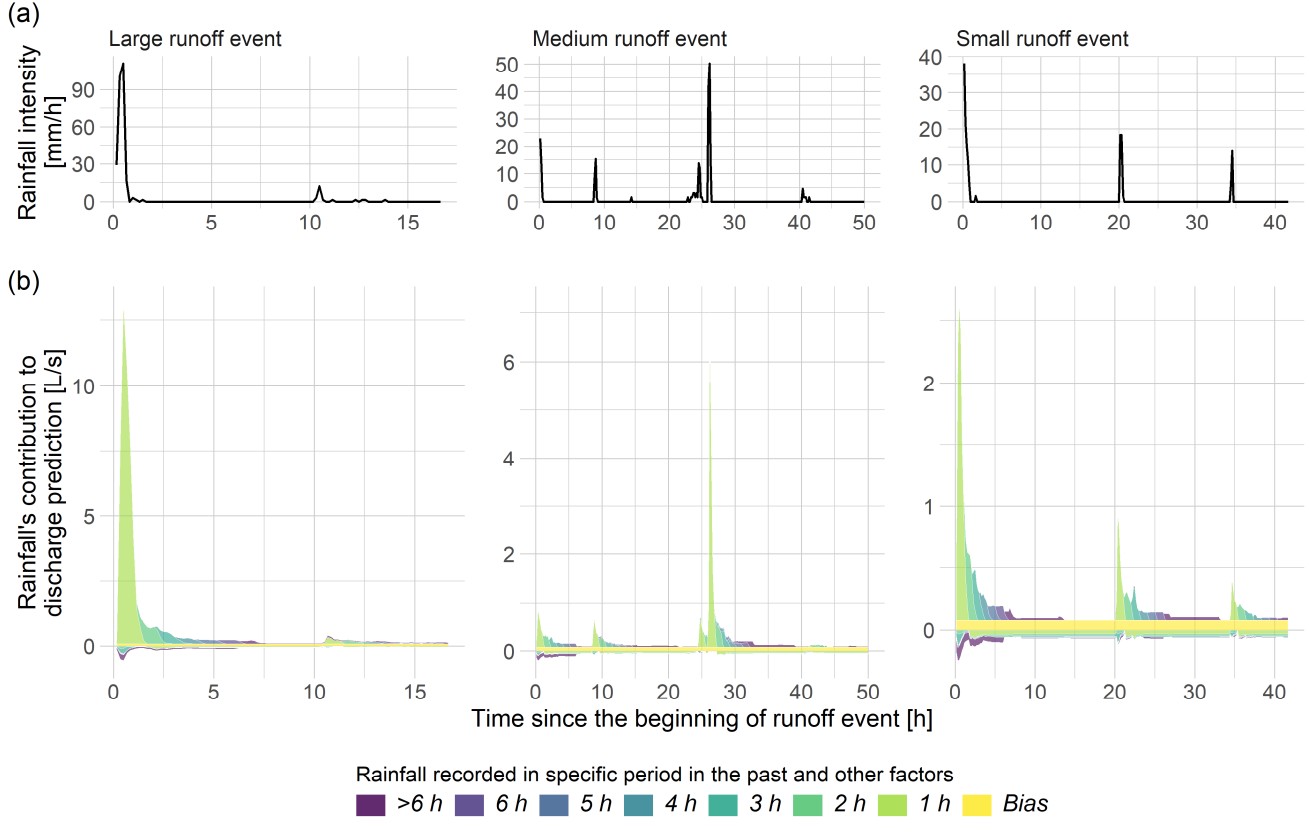

**Figure 9 (a) Rainfall time series and (b) decomposed hydrographs of a large, medium, and small runoff event from the Washington Street SuDS site (WS). The model used to derive the hydrographs was built using aggregation option 1 to create the rainfall depth features in the outer cross-validation iteration 1. Observational SHAP values were used in the computation.**

Fig. 10 shows the average age of the rainfall that controls the discharge predictions of a large, medium, and small runoff event in a WS model, which was computed using Eq. (12). The average rainfall age generally decreases when rainfall occurs and increases when rainfall stops. The average age is affected by rainfall depth. For instance, in the large runoff event, the average age of the rainfall around the 11th hour only decreases to approximately 2.5 h due to the small size of the rainfall compared with the large rainfalls observed at the beginning of the event. This result implies that large rainfalls generally affect the runoff generation process for longer periods than small rainfalls, which is reasonable. The average age computation method defined in Eq. (12) is similar to the end-member mixing model used in hydrology to determine the runoff sources (Burns et al., 2001). The average ages computed in this study may also be useful for determining the runoff sources. Considering the small size of the WS (0.0011 km²), the runoffs associated with an average age older than 1 h may be partly contributed by stormwater that slowly passes through the soil layers of green roofs and bioretention cells.

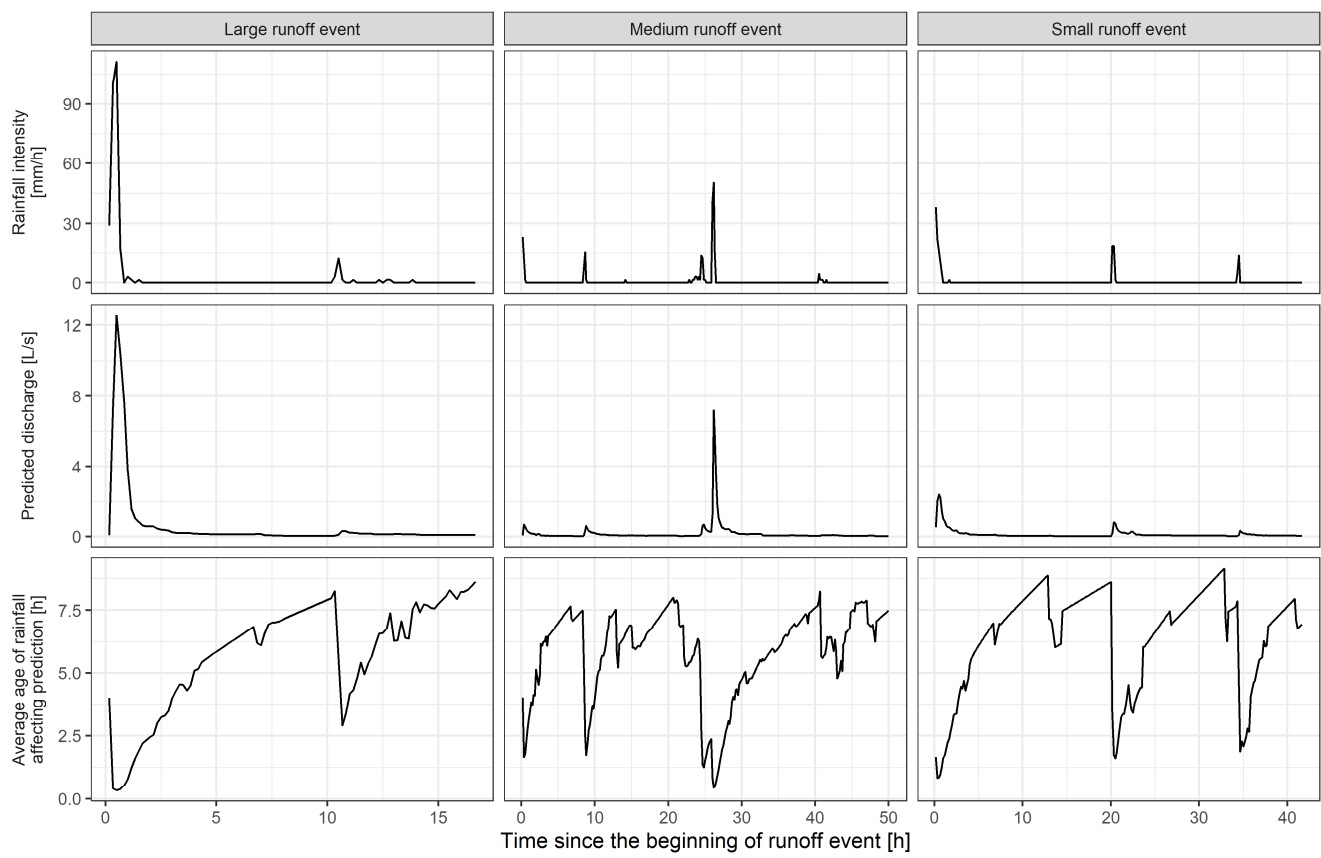


**Figure 10 Rainfall time series, predicted hydrographs, and average age of the rainfalls controlling the runoff prediction. The model used here was built using aggregation option 1 to create the rainfall depth features in the outer cross-validation iteration 1. The observational SHAP values were used in the computation.**

This study, however, does not compare the hydrograph decomposition and average age of the rainfalls derived for different runoff events using different models. A systematic multi-event and multi-model comparison is therefore left for future research. It would also be meaningful to compare these quantities derived using the proposed methods against those from other commonly used approaches in process-based modeling and tracer and isotope hydrology.

## 4 Conclusions

This study proposes an explainable machine learning method to predict and interpret the hydrological responses of sustainable urban drainage systems (SuDS) to rainfalls at sub-hourly time scales. The proposed methods are applied to two SuDS catchments of different sizes, SuDS practice types, and data availabilities to predict discharge and produce models with good prediction accuracies ($NSE > 0.7$ and $R^2 > 0.7$ for all models). These models consistently outperform linear regression models and have similar prediction accuracies to a process-based model for one study site. The local feature attribution methods, i.e., the SHAP values, are then adopted to explain the basis of each discharge prediction. The contribution of the

rainfall of each time step to each discharge prediction is further derived. The average contribution of rainfall at different past time steps to the discharge prediction is then calculated using all of the input samples and used as a feature importance measure. The feature importance measure is useful for analyzing the model structure and provides insights into the runoff generation process. In all of the machine learning models, the discharge prediction is found be substantially affected by the rainfalls in the past few time steps, which is reasonable considering the small catchment sizes. In models trained using only one month of

data, fluctuations are detected in the importance derived for the rainfalls across different time steps. The unrealistic fluctuation patterns reveal the insufficiencies of the model structures. This study further combines the explanations of low- and high-flow predictions to understand the model behavior for input samples from different regions of the input domain. A conceptual rainfall–runoff model is identified in these models, in which the final discharge prediction is the sum of the basic prediction and contributions associated with the rainfalls of different time steps. However, the rainfall contributions can be negative,

which distinguishes this representation from commonly used processed-based models. This study briefly shows that a discharge prediction can be decomposed into the contributions of rainfall at each time step, which can be useful for hydrograph separation and determining the runoff source.

It was difficult to build process-based hydrological models for the two SuDS catchments examined in this study due to insufficient information regarding the physical properties and drainage processes. This study designs a simple feature

engineering method to facilitate the application of the commonly used machine learning algorithms to model rainfall–runoff correlations at sub-hourly time scales. In this method, rainfall depth features extracted from high-resolution rainfall time series are used as input variables of the machine learning models, instead of the original time series. The depth features are calculated as the rainfall depths measured over specific time intervals, the number and locations of which are controlled by a few hyperparameters. Bayesian optimization methods and a nested cross-validation procedure are adopted to derive the optimal

feature engineering hyperparameters along with the hyperparameters required by the specific machine learning algorithms. The model training process is automatic and only requires that the range of possible hyperparameter values be defined. In this study, the hyperparameter ranges are set to be the same for the two study sites. The proposed methods are also tested on a few other SuDS catchments with different configurations, and the prediction accuracies are generally found to be satisfactory. The source code used in this research and the additional cases studies can be found in the Code availability section.

Local feature attribution methods are used in this study to explain the basis of each prediction. These explanations are found to be useful for checking the validity of the machine learning models, i.e., whether the input–output correlations learned by the models make sense for the various input samples. The feature importance measure derived from the local explanations is compared with commonly used global feature importance measures and found to generate similar results. The local explanations are also used in applications such as the development of conceptual rainfall runoff models, hydrograph separation,

and the identification of runoff sources. Overall, the application of local feature attribution methods increases the transparency of machine learning predictions. These methods can be useful for detecting errors in the model structures and inferring the hydrological processes being modeled from the data.

Future research may proceed in the following directions. First, this study designed a feature engineering method to reduce the dimension of input variables, i.e., rainfall time series. However, feature engineering methods can be designed arbitrarily and it can be computationally expensive to identify the optimal methods. Future studies can thus explore the application of machine learning methods that are specifically designed for modeling high-dimensional time series data, such as the long short-term memory (LSTM) networks in deep learning. Second, the methods proposed in this study are only applied to model the correlations between rainfall time series and discharge in a few U.S. sites. The proposed methods should therefore be tested in more sites worldwide to model the correlations between other variables, although this may require the development of new feature engineering methods. Finally, it may be useful to compare the hydrological processes inferred by the machine learning models to those obtained using conventional methods. For instance, the hydrograph decomposition methods presented in this study can be compared with commonly used process-based modeling approaches.

**Code availability**

The source code used in this study is available at https://github.com/stsfk/ExplainableML_SuDS, where a few example applications are also presented. The following R packages are used for modeling and analysis in this research: xgboost (Chen and He, 2020), tidymodels (Kuhn and Wickham, 2020), lubridate (Grolemund and Wickham, 2011), RcppRoll (Ushey, 2018), zeallot (Teetor, 2018), ParBayesianOptimization (Wilson, 2021), and hydroGOF (Zambrano-Bigiarini, 2020). The following Python packages are used: shap (Lundberg and Lee, 2017), NumPy (Harris et al., 2020), and xgboost (Chen and Guestrin, 2016). All the R and Python package packages used in this research are freely available online.

**Data availability**

The data of the two study sites examined in this study is obtained from the United States Geological Survey (USGS), Clermont County, Ohio, U.S., and the United States Environmental Protection Agency (US EPA). The identification numbers of the USGS monitoring sites for the Washington Street SuDS site (WS) are 412533081221500, 412535081221400, and 412535081221402. The data of the Shayler Crossing Watershed can be downloaded at https://doi.org/10.23719/1378947. The SWMM model used in this research is developed in Lee et al. (2018a).

**Author contribution**

YY designed the study, acquired the data, wrote the code, conducted the numerical experiment, analyzed the results, and prepared the manuscript. TFMC contributed to the design of numerical experiments, supervised the study, validated the results, and revised the manuscript.

## Competing interests

The authors declare that they have no conflict of interest.

## Acknowledgments

The work described in this paper was partly supported by a grant from the Research Grants Council of the Hong Kong Special Administrative Region, China (Project No. HKU17255516), and partly supported by the RGC Theme-based Research Scheme (Grant No: T21-711/16-R) funded by the Research Grants Council of the Hong Kong Special Administrative Region, China. We thank Robert Darner from USGS, Christ Nietch from USEPA, Joong Gwang Lee from Center for Urban Green Infrastructure Engineering, and Bill Mellman from Clermont County Water Resources for providing data for this research.

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
