# Peer review of "Modeling and interpreting hydrological responses of sustainable urban drainage systems with explainable machine learning methods"

_Hydrology and Earth System Sciences, 2020_

## Referee Comment (RC1) · Anonymous Referee #1 · 14 Oct 2020

In General Firstly, this work is innovative for explaining machine learning predictions in hydrology forecasting. With applying AI in various fields and getting excellent results, it is a hot topic to interpret the machine learning. But this manuscript still has some questions needed revised. Generally, it is a good research point, but manuscript is hard to understand. The logic of this paper is not clear that I cannot figure out what information explained by SHAP model and what relationship of hydrological response and selected hyperparameters. I think the main question is limited input variables (only Rainfall depth). I cannot agree that the design rainfall depth features (Section

2.1.1) reflect SuDS hydrological process. Thus, the hyperparameters of m, l, q, account_CumRain and account_season have little meaning for interpreting hydrological process in SuDS. Originally, SHAP is a game theoretic approach to explain the output of machine learning model. So maybe more physical observation variables are needed to selected as input variables. Therefore, I suggest this manuscript for Major Revision and Resubmission. Point 1:Whether the constructed data feature mining algorithm corresponds to the reference standard in the folded data part? Point 2: "The framework is particularly useful for urban catchments where the information for setting up process-based models is insufficient."ïijĹline580ïijĽIs this statement reasonable? Do similar expressions still exist in the full text? Point 3:Adding quantitative analysis to the conclusion section should be more convincing. Point 4:Compared with the commonly used urban rainfall runoff models, what are the obvious advantages of this model?

Specially Line 620-780: It is difficult for finding the references because of improperly format. Line 9: How do you define the "fine temporal scales"? It is an important concept in your forecasting, but it is not clear. Line 131: Why you use Dt-a,t-b for aggregating rainfall depth? In Line 84 said many observation data became available, but why only the rainfall data? Do you have other data? Line 6-14 and Line 560-595: In the section of abstract and conclusion, the quantitative results are absent and the qualitative descriptions are not enough.

---

## Short Comment (SC1) · 16 Oct 2020

We would like to thank the reviewer for providing insightful comments for improving the quality of the paper. Our short response to the comments is as follow, which outlines our plans to revise the paper.

1. We agree with the reviewer that the connections between the SHAP values of the hyperparameters and the hydrological processes are hard to understand. In the original submission, in addition to the rainfall-runoff machine learning models, we built

models to predict the accuracy of rainfall-runoff models using their feature engineering hyperparameter values. And then identified the impact of each hyperparameter on the accuracy of rainfall-runoff models using SHAP. The premise is that the hyperparameters control what information of the rainfall to be passed into the rainfall-runoff models. Thus, if the information to be passed resulted in models with high prediction accuracy, we consider such information to be relevant to the runoff generation processes. However, we found this connection to be indirect and affected by the learning ability of the rainfall-runoff machine learning models. We also did not explain the connections clearly in the original submission. Thus, we plan to remove or significantly shorten the content on this point.

The SHAP method is also used to explain the basis of the predictions made by the rainfall-runoff machine learning models, i.e., the impact of rainfall at each time steps to runoffs at subsequent steps were quantified. Such information was then used for determining catchment response time and hydrograph separation (see Figures 9-11, line 475 to 530). As stated in line 497 to 498, "As SHAP values satisfy local accuracy property, it is possible to decompose a flow rate prediction into flow rates contributed by each feature." The SHAP values of rainfall depth features can be further attributed to each rainfall depth record, which measures its contribution to a rainfall record. We understand that the other hydrometeorological variables are not considered, the rainfall-runoff correlation presented in the models is only a simplified representation of the true processes. As the models have good prediction accuracies, we may say that the models learned a good approximation of the major processes in the real-world. We will further discuss this point in the updated manuscript and remind the readers that this is a modeling result and further verifications are needed.

2. The reviewer suggests that more physical observation variables could be considered, as they are influencing factors of the hydrological processes. We agree with this point. Unfortunately, we do not have other physical observation data, i.e., only rainfall and runoff time series are available. The design configurations of the drainage

systems of the first study site are also unavailable. Thus, it is very difficult, if not impossible, to set up process-based models. The lack of data is also a motivation for using machine learning methods. To address this shortcoming, we included two additional hyperparameters, account_season and account_CumRain, to let the machine learning method explore whether there are seasonality or changes in the long-term performance of the sustainable drainage systems. We will further clarify this point in the updated manuscript.

3. We claimed that our framework is useful for catchments with insufficient data for setting up process-based models. However, more justifications could be provided, e.g., our method does not require the measurements of the physical properties of the catchment but does require rainfall-runoff time series. We will add more explanations.

4. The proposed feature engineering algorithm corresponds to the method to derive rainfall depth features (Dt-a,t-b) and the nested cross-validation procedures to select the optimal set of features and the hyperparameters of machine learning methods. To make the logic of the section 2 clearer, we will add a short summary before introducing the detailed methods.

5. The "fine-temporal scale" refers to "sub-hourly". We will use this more specific term throughout the paper.

Modeling the rainfall-runoff correlation at a sub-hourly scale (i.e., 10-min) is quite difficult, as the dimension of the input variable ("rainfall") could be a few thousand. To solve this problem, we propose a feature engineering method to reduce the dimension of the input variable and a method to select the optimal features.

6. The use of Dt-a,t-b: we mentioned the meaning of using them in line 155, "The collection of Dt-a,t-b are a compact representation of the original rainfall time series. The resolution and the encoded information of this representation are controlled by m, l, and q." The other reasons for using them include the preservation of the temporal distribution of the rainfall and the direct connection to the raw rainfall input. However,

as the reviewer pointed out, the reason for using Dt-a,t-b is presented after it has been defined, which makes it hard to understand. We plan to rewrite some sections to improve the readability of this paper, especially for the method section.

7. We will add quantitative summaries of the experiments to the conclusion section, as suggested by the reviewer. Qualitative descriptions will also be updated, e.g., the reasons for using machine learning models over the conventional process-based model and shortcomings of not including other hydrometeorological variables.

8. The advantage of the proposed method to the conventional processed-based method is that it does not require measurements of the physical properties of the catchment or assumptions regarding the hydrological processes. Basically, it provides an opportunity for finding statistical connections between two variables without knowing the underlying processes. We present the difficulties in applying process-based models in line 42 to 56. The challenges of using process-based models for the two study sites are lack of information on the physical properties of the catchment and complexities in parameterizing the drainage systems, as discussed in line 279 to 283 and line 323 to 324. However, we regret have not discussed it again in the results and the conclusion section.

The disadvantage of machine learning models is that they are hard to interpret or lack of transparency, i.e., we do not know why a certain prediction is made. Thus, in this research, we propose to use SHAP to analyze the basis of each prediction. The other disadvantage is that they are difficult to model high-dimensional input-output correlation. Thus, a new feature engineering method for reducing the dimension of the input variable is proposed in this research.

9. The cited literature is formatted automatically using a reference management software with the HESS style selected, and we are sorry for the mistakes in the format. We will check the citation style manually to avoid mistakes. We also plan to update the structure of the manuscript to make it more concise and easier to understand and hire

a professional English editor to correct grammar mistakes.

In summary, we understand the reviewer's concerns on only using rainfall time series as the input variable. However, the other hydrometeorological variables for the two study sites are unavailable. The runoff responses to storms of the two catchments are investigated in this study, which is useful for understanding the effectiveness of sustainable urban drainage systems. As the two catchments are very small (1,000 m2 and 1 km2) and good prediction accuracies are achieved by machine learning models, we may consider the models have learned a reasonably good approximation of the runoff-generation processes (see our response #1). Does the reviewer suggest additional analysis or case studies? We would highly appreciate it if the reviewer could further comment on our revision plan or provide additional feedback.

Any comments on the manuscript or this response letter from anyone are appreciated.

---

## Referee Comment (RC3) · Georgia Papacharalampous (Referee) · 5 Nov 2020

**Review Report**

| | |
|---|---|
| Journal: | Hydrology and Earth System Sciences |
| Manuscript's Ref.: | hess-2020-460 |
| Title: | Modeling and interpreting hydrological responses of sustainable urban drainage systems with explainable machine learning methods |
| Authors: | Yang Yang, Ting Fong May Chui |
| Reviewer: | Georgia Papacharalampous |
| Date agreed: | 2020-10-09 |
| Date submitted: | 2020-11-05 |
| Recommendation: | Minor revisions |

**Summary**

The paper focuses on the predictive modeling of sustainable drainage systems (SuDS) at fine temporal scales using boosting (Friedman 2001). Several boosting variants are formed and exploited in two case studies, while comparisons with the linear regression algorithm and the Storm Water Management Model (SWMM; Rossman 2015) are also provided. Furthermore, the SHapley Additive exPlanations (SHAP) method (Lundberg and Lee 2017) is used to explain the contribution of each variable (else referred to as "feature") to the issued predictions, thereby facilitating interpretability to some extent.

**General comments**

In general, I find that the manuscript is well-formulated and -written, and I think that the work done so far (including the release of the R codes at GitHub) should be appreciated. Nonetheless, I also think that there is some room for improvement before publication.

I recommend minor revisions. My comments are given right below.

**Comments**

(1) To my view, the following clarification is required: Which are the similarities and differences between basic variable importance measures (available in the `xgboost` R package) and the SHAP methodology (available in the `SHAPforxgboost R` package)?

(2)    Since interpretability is one of the main themes of the present work, I feel that a comparison (direct or indirect, depending on the answer to comment #1) between basic variable importance measures and the SHAP methodology is currently missing from the manuscript and should be necessarily made for both case studies. New computations are needed for this comment to be fully addressed (independently of the answer to comment #1); however, these computations will only require the `xgboost R` package (which is already used in the paper).

(3)    In light of comments #1 and #2, other hydrological studies using boosting or random forests while also emphasizing on interpretability (by using variable importance measures) could be discussed (in comparison to the present study) somewhere in the manuscript. What is the added value of the present work with respect to such existing works?

(4)    In the "Introduction" section, it is written that "only a few studies adopted machine learning methods to investigate the hydrological processes of SuDS", with the studies by Eric et al. (2015), Khan et al. (2013), Li et al. (2019), and Yang and Chui (2019) being discussed as examples of such studies. Since such studies are quite close to the present work, more of them could be reported (provided that they exist).

(5)    In the same section, it is also written that "modeling the responses of SuDS at fine temporal scales requires high-dimensional hydrometeorological time series to be used as input, which is difficult in machine learning". Could this sentence be further elaborated? I would say that the opposite holds, i.e., that machine learning methods are ideal for handling high-dimensional hydrometeorological time series.

(6)    The reader could also be referred to several specialized books (e.g., Hastie et al. 2009; James et al. 2013; Witten et al. 2007), for further information on the machine learning (or statistical learning) methods used in the paper.

(7)    Another concern of mine is related to the small number of real-world cases examined in the paper. I think that the application of the proposed procedures to large real-word datasets (comprising hundreds of cases) should be addressed at least with extensive relevant discussions in the manuscript (e.g., future research recommendations). (Currently, it is only suggested using "the SHAP method in more case studies"). To my view, these extensive discussions are important, especially given that (i) there are studies in the hydro-meteorological literature validating their

models using big datasets, and (ii) the first aim of the paper is to "evaluate the usefulness of machine learning methods in predicting the hydrological responses of SuDS at fine temporal scales". The necessity of evaluating machine learning methods using big datasets is extensively discussed by Boulesteix et al. (2018).

(8)    In the "Conclusions" section it is written that "the proposed model training methods are semi-automatic, requiring minimal user input". It would be useful to discuss (somewhere in the paper) which parts of the proposed methods are not (fully) automatic, and how one could overcome this limitation to allow large-scale (even global-scale) investigations (see also comment #7).

(9)    Currently, the use of the `xgboost` and `SHAPforxgboost` R packages is reported in the manuscript. To my view, all utilized software packages (which, of course, at the moment can be found online at https://github.com/stsfk/explainable_ml_hydro, since the R code has been made available) should necessarily be reported and cited in the paper.

(10)   Finally, the manuscript is not typo-free at the moment. Particular attention should be placed on the mathematical notations. For instance, the transpose operator should not be written in italics (therefore, $T$ should be replaced with T) and the vectors should be bolded (therefore, $X_{t-m,t}$ should be replaced with $\boldsymbol{X}_{t-m,t}$).

**References**

Boulesteix AL, Binder H, Abrahamowicz M, Sauerbrei W, for the Simulation Panel of the STRATOS Initiative (2018) On the necessity and design of studies comparing statistical methods. Biometrical Journal 60(1):216–218. doi:10.1002/bimj.201700129

Hastie T, Tibshirani R, Friedman JH (2009) The elements of statistical learning: Data mining, inference and prediction, second edition. Springer, New York. doi:10.1007/978-0-387-84858-7

James G, Witten D, Hastie T, Tibshirani R (2013) An introduction to statistical learning. Springer, New York. doi:10.1007/978-1-4614-7138-7

Witten IH, Frank E, Hall MA, Pal CJ (2017) Data Mining: Practical machine learning tools and techniques, fourth edition. Elsevier Inc. ISBN:978-0-12-804291-5

---

## Author Comment (AC1) · 28 Dec 2020

We would like to thank the Anonymous Referee #1 for providing valuable and constructive suggestions regarding our manuscript. Please find our responses below.

*Italicized text: comments made by Referee #1.*
Blue text: Authors' responses.

*1. Firstly, this work is innovative for explaining machine learning predictions in hydrology forecasting. With applying AI in various fields and getting excellent results, it is a hot topic to interpret the machine learning. But this manuscript still has some questions needed revised. Generally, it is a good research point, but manuscript is hard to understand.*

The referee commented that our manuscript is hard to understand. To improve the readability of the paper, we updated the manuscript in the following aspects: (a) simplification of the methods, (b) removal of non-essential findings, and (c) re-organizing the paper according to the updated research objective. The details are as follow.

(a) Simplification of the methods. We updated the feature engineering method and the hyperparameter optimization method, both of which are used for training machine learning models.

In the revised manuscript, the high-resolution rainfall time series is converted into rainfall depth features using three hyperparameters, $m$, $l$, and $n$. Only the rainfalls recorded between $t - m$ and $t - 0$ are considered. Each rainfall depth recorded between $t - l$ and $t - 0$ is used for creating rainfall depth features. And $n$ intervals are created for aggregating the rainfall recorded between $t - l - 1$ and $t - m$, and the intervals roughly form an arithmetic sequence. See the illustration in Figure 1. The updated method is easier to understand, and the complex equations (Eqs. 5-7) in the original submission can be removed.

[Figure]

Figure 1. Illustration of the method to derive rainfall depth features. $p_{t-i}$ denotes the rainfall depth recorded at time $t - i$.

We also simplified the hyperparameter optimization method. In particular, in the revised manuscript, we no longer differentiate the feature engineering hyperparameters and the

XGBoost hyperparameters, and all the hyperparameters were optimized together through an automated Bayesian optimization method (Snoek et al., 2012). Figure 2 shows that the Bayesian optimization method can find high-quality solutions (as indicated by low inner cross-validation (CV) errors) in a few optimization steps.

[Figure]

Figure 2. (a) The model's prediction accuracy associated with the hyperparameters evaluated during each optimization step. The prediction accuracy is measured by the root-mean-square error (RMSE) of the predictions obtained during the inner cross-validation (CV) iterations. (b) The expected utility of the candidate hyperparameters evaluated during each optimization step.

We also investigated whether the optimization method resulted in overfitting the model selection criterion. As indicated by the positive correlation between the inner and outer CV errors in Figure 3 (i.e., the good models found during the optimization process also had good performances during testing), the optimization method did not overfit the model selection criterion. Readers do not need to understand the technical details regarding Bayesian optimization, and the method is described briefly in the revised manuscript. In this way, the descriptions on the resampling scheme, the choice of feature engineering and XGBoost hyperparameters, and the model selection method can all be removed or substantially shortened. The models derived using the updated and the old methods were found to have comparable prediction accuracies.

[Figure]

Figure 3. The model's prediction errors estimated during the inner and the outer cross-validation (CV) iterations for each set of candidate hyperparameters evaluated during the optimization. The prediction error is measured by root-mean-square error (RMSE). Each subfigure shows the result obtained for a rainfall depth feature aggregation method and during an outer CV iteration.

(b) Removal of non-essential findings. In the updated manuscript, the results of XGBoost hyperparameters optimization (Section 3.1) were removed due to the updated method and the new research objective. The results on interpreting the feature engineering hyperparameters (Section 3.3.1) were also removed due to their indirect connections to the hydrological processes. The descriptions of goodness-of-fit of the trained models (Section 3.2) were shortened.

(c) Re-organizing the paper according to the new research objective. We removed the following research objective from the updated manuscript, "develop and present tools and methods for building higher quality machine learning models for SuDS-related studies and demonstrate the applications". And the overall objective became "modelling the hydrological responses of SuDS to rainfalls and examining the basis of predictions using interpretation methods". Therefore, we removed the "demonstration" element from the original submission, and focused on developing methods for modelling SuDS and interpreting the model predictions. New

findings applying the proposed methods were also reported. The following changes were made in terms of the content of the paper.

In the introduction section of the updated manuscript, an introduction to the interpretation methods and their applications in hydrology is presented. This will help readers understand the research objectives.

In the methods section, we extended the introduction to the interpretation methods. In particular, the differences between the local and the global interpretation methods (such as the commonly used gain and cover metrics for XGBoost) were discussed, and the two methods (the observational and the conditional methods) to compute the SHAP values and their implications were introduced. We also substantially shortened the description of machine learning model training methods. More focus was given to the interpretation methods, as literature on this topic is currently lacking in the field of hydrology, and readers can get a better understanding of the results if more information on the interpretation methods is provided.

In the results section, we removed some of the results on machine learning model training and the results on interpreting feature engineering hyperparameters. The comparison between SHAP and other global interpretation methods was added, as suggested by Referee #2. We also examined the differences in explanations when using different interpretation methods for different models. In addition, we explained the potential applications of the interpretation methods in greater detail.

*2. The logic of this paper is not clear that I cannot figure out what information explained by SHAP model and what relationship of hydrological response and selected hyperparameters.*

The SHAP method is a feature attribution method, i.e., it aims to explain how much each feature contributed to the output of a model for a particular sample (Janzing et al., 2019). For this particular study, the rainfall's contribution to the subsequent runoff predictions at each time step is computed. In the updated manuscript, we added a formal introduction to feature attribution problem and their methods. We hope this can help the readers understand the results.

In response to this particular comment, we removed the content on explaining the hyperparameters in the updated manuscript, as we found their connections to the hydrological processes are indirect. The updated manuscript only explains the contribution of rainfall to runoff predictions at different time steps.

*3. I think the main question is limited input variables (only Rainfall depth). I cannot agree that the design rainfall depth features (Section 2.1.1) reflect SuDS hydrological process. Thus, the hyperparameters of m, l, q, account_CumRain and account_season have little meaning for interpreting hydrological process in SuDS. Originally, SHAP is a game theoretic approach to explain the output of machine learning model. So maybe more physical observation variables are needed to selected as input variables. Therefore, I suggest this manuscript for Major Revision and Resubmission.*

We agree with this comment that more variables can be included in the machine learning models. However, for the study site, only the rainfall and the runoff time series were available. The lack of data (such as the physical properties of the catchment) for setting up process-based models was also a motivation for using machine learning methods. In addition, this study focuses on modelling stormwater runoffs of small-scale urban drainage infrastructures during the wet period (i.e., within 24 hours of rainfall events), thus rainfall is the main driver of the system being modelled. Studying how a runoff prediction is affected by the rainfall of each time step is meaningful. Moreover, we also considered additional input features to the machine learning models to account for potential seasonality of the performance of the SuDS. Finally, this study recommends future studies to include more variables in machine learning models in the conclusion section.

In response to this comment, we removed the content on explaining the hyperparameters, as they are indirectly connected to the hydrological processes of SuDS.

In the updated manuscript, we presented SHAP as a method to explain the basis of a prediction for checking whether that prediction can be trusted. That is, we do not think the hydrological processes inferred by machine learning models are necessarily true. We further clarified this point by adding studies comparing the inferred hydrological processes of different machine learning models derived using different methods for computing SHAP values. As an example, Figure 4 shows that different machine learning models considered the rainfall's contributions to runoff predictions differently. Thus, there were considerable uncertainties in interpreting machine learning model predictions. The existence of various possible explanations was referred to as equifinality in Schmidt et al. (2020), which is an important concept in hydrological modelling (Beven and Freer, 2001). We reported this issue in the updated manuscript.

[Figure]

Figure 4. Rainfall's contribution to runoff prediction at different time steps ahead. Each subfigure shows the results obtained in different outer cross-validation (CV) iterations and the mean values derived using a feature engineering method and a SHAP computation method.

*4. Point 1:Whether the constructed data feature mining algorithm corresponds to the reference standard in the folded data part?*

In the updated manuscript, we no longer discuss the specific values of feature engineering hyperparameters due to their indirect connections to the hydrological processes. We used a Bayesian optimization method to optimize the hyperparameters automatically.

*5. Point 2: "The framework is particularly useful for urban catchments where the information for setting up process-based models is insufficient." Is this statement reasonable? Do similar expressions still exist in the full text?*

Thank you for raising this question. We think further clarification is needed in the updated manuscript. We claimed that "the framework is particularly useful when information for setting up process-based models is insufficient". Here, the information refers to the knowledge about the physical properties and the physical processes of the system being modelled. This point is demonstrated using two case studies. The physical properties of the drainage systems in the first case study were unknown, and it was also difficult to represent the unknown leakage from

SuDS using process-based models. The second case study site contains a large-scale and complex drainage network, requiring many parameters to characterize their physical properties in process-based models. Machine learning models were built relatively easily in the two case studies and showed relatively high prediction accuracies. However, we argue that process-based hydrological models are useful for examining the involved hydrological processes.

Therefore, machine learning methods are useful for modelling the statistical correlations between interested random variables of the catchment when observation data of the variables are available, and the trained machine learning models can serve as a baseline for evaluating process-based models. We modified the original statement to match this conclusion in the updated manuscript.

*6. Point 3:Adding quantitative analysis to the conclusion section should be more convincing.*

In the updated manuscript, we added the performance metrics of the trained machine learning models in the abstract and the conclusion section.

*7. Point 4:Compared with the commonly used urban rainfall runoff models, what are the obvious advantages of this model?*

The advantage of using machine learning methods is that they only require observation data of the interested random variables and do not require the involved physical processes to be characterized. Machine learning models can generally be set up easily and can potentially provide high-quality predictions. Machine learning models may be used as baseline models for evaluation of process-based models. More explanations are provided in our response to comment #5.

*8. Line 620-780: It is difficult for finding the references because of improperly format.*

Thank you for catching the errors. We updated the citation styles throughout the manuscript to meet HESS requirement.

*9. Line 9: How do you define the "fine temporal scales"? It is an important concept in your forecasting, but it is not clear.*

"Fine temporal scales" refers to sub-hourly scales. In the updated manuscript, the term "sub-hourly scales" is used as it is more specific.

*10. Line 131: Why you use Dt-a,t-b for aggregating rainfall depth?*

The lower-dimensional rainfall depth features $D_{t-a,t-b}$ were used because the dimension of the original rainfall time series can be very high (e.g., 1,000 time steps), and some machine learning methods have difficulties to learn the high-dimensional correlation between the input and the output random variables. Thus, we designed a feature engineering method to lower the dimension of the input variables of the machine learning models, and the number of features used is controlled by three hyperparameters. In fact, the feature engineering method allows the rainfall depth features to be very similar to the original time series. And the values of the hyperparameters are chosen according to the prediction accuracy of the resulted machine learning models. This point is explained in the updated manuscript.

*10. In Line 84 said many observation data became available, but why only the rainfall data? Do you have other data?*

We only have rainfall and runoff data for both study sites, as reported in our response to comment #3. The sentence "more observation data became available" was referring to the fact that the rainfall and runoff are being monitored in more SuDS sites globally. However, as pointed out by Schaffitel et al. (2020), monitoring data of other variables concerning urban hydrology are still currently lacking. Therefore, it can be useful to present a study that focuses only on the correlation between rainfall and runoff time series. More information on this issue is also presented in our response to comment #3. We commented on this issue in the updated manuscript.

*11. Line 6-14 and Line 560-595: In the section of abstract and conclusion, the quantitative results are absent and the qualitative descriptions are not enough.*

Quantitative results are presented in the updated manuscript.

**Reference**

Beven, K. and Freer, J.: Equifinality, data assimilation, and uncertainty estimation in mechanistic modelling of complex environmental systems using the GLUE methodology, J. Hydrol., 249(1–4), 11–29, https://doi.org/10.1016/S0022-1694(01)00421-8, 2001.

Janzing, D., Minorics, L. and Blöbaum, P.: Feature relevance quantification in explainable AI: A causal problem, https://arxiv.org/abs/1910.13413, 2019.

Schaffitel, A., Schuetz, T. and Weiler, M.: A distributed soil moisture, temperature and infiltrometer dataset for permeable pavements and green spaces, Earth Syst. Sci. Data, 12(1), 501–517, https://doi.org/10.5194/essd-12-501-2020, 2020.

Schmidt, L., Heße, F., Attinger, S. and Kumar, R.: Challenges in Applying Machine Learning Models for Hydrological Inference: A Case Study for Flooding Events Across Germany, Water Resour. Res., 56(5), https://doi.org/10.1029/2019WR025924, 2020.

Snoek, J., Larochelle, H. and Adams, R. P.: Practical Bayesian optimization of machine learning algorithms, in Advances in Neural Information Processing Systems, vol. 4, pp. 2951–2959, https://arxiv.org/abs/1206.2944v2, 2012.

---

## Author Comment (AC2) · 28 Dec 2020

We would like to thank Dr. Georgia A Papacharalampous (Referee #2) for providing insightful comments for improving our paper. Our responses to her comments are as follow.

*Italicized text: comments made by Dr. Georgia A Papacharalampous (Referee #2).*
Blue text: Authors' responses.

**Summary***: The paper focuses on the predictive modeling of sustainable drainage systems (SuDS) at fine temporal scales using boosting (Friedman 2001). Several boosting variants are formed and exploited in two case studies, while comparisons with the linear regression algorithm and the Storm Water Management Model (SWMM; Rossman 2015) are also provided. Furthermore, the SHapley Additive exPlanations (SHAP) method (Lundberg and Lee 2017) is used to explain the contribution of each variable (else referred to as "feature") to the issued predictions, thereby facilitating interpretability to some extent.*

Thank you for providing a nice summary of our research.

**General comments:** *In general, I find that the manuscript is well-formulated and -written, and I think that the work done so far (including the release of the R codes at GitHub) should be appreciated. Nonetheless, I also think that there is some room for improvement before publication.*

*I recommend minor revisions. My comments are given right below.*

Thank you for the positive assessment. We revised the manuscripts according to your comments. In particular, we added more discussions on the machine learning model interpretation methods and a comparison between the explanations derived using different interpretation methods. We also plan to improve the documentation of the source code on GitHub when submitting the revision. Please find our responses and some of the new results below.

In addition, in response to the comments provided by Anonymous Referee #1, we improved the readability of the paper by making the following changes: (a) simplification of the methods, (b) removal of non-essential findings, and (c) re-organizing the paper according to the new research objective. The modifications did not change the overall content and the conclusion of the paper. Detailed information can be found in Author Comment #1 (AC1).

***Comments:***

*(1) To my view, the following clarification is required: Which are the similarities and differences between basic variable importance measures (available in the xgboost R package) and the SHAP methodology (available in the SHAPforxgboost R package)?*

The main differences between SHAP and the basic importance measures (such as gain and cover) are as follow.

(a) SHAP is a model-agnostic interpretation methods, and the other importance measures provided by the "xgboost::xgb.importance" function in R package are model-specific (Chen et al., 2020). The advantage of model-agnostic methods is that they can be applied to various machine learning models and thereby allow comparisons between different types of models in terms of the derived interpretations (Ribeiro et al., 2016).

(b) SHAP is a local interpretation method while the other methods provided by the "xgboost::xgb.importance" function are global interpretation methods. The local interpretation methods are designed for interpreting the prediction made for individual input samples, and the global methods are independent of the input samples and often explain the structure of the model (Lundberg and Lee, 2016). Therefore, in this study, SHAP can be used to analyse a specific runoff prediction, and the other methods cannot be used for this task.

(c) Theoretically, SHAP is the only method that provides interpretations that satisfy a series of desired properties, such as local accuracy, missingness, and consistency (Lundberg et al., 2020).

In the updated manuscript, we added discussions regarding the differences between various model interpretation methods.

*(2) Since interpretability is one of the main themes of the present work, I feel that a comparison (direct or indirect, depending on the answer to comment #1) between basic variable importance measures and the SHAP methodology is currently missing from the manuscript and should be necessarily made for both case studies. New computations are needed for this comment to be fully addressed (independently of the answer to comment #1); however, these computations will only require the xgboost R package (which is already used in the paper).*

Thank you for your suggestion. In the updated manuscript, we added a comparison between feature importance derived using different interpretation methods. As shown in Figure 1, for each machine learning model, the rainfall's contribution to the runoff prediction at each time step was computed using different methods. As we explained in the response to comment #1, the gain, the cover, and the frequency are all global interpretation methods, it is not possible to

compute the importance of the rainfall for a specific input sample. Thus, each dashed line in Figure 1 corresponds to the results obtained for a model for all input samples. In general, we found that gain and SHAP offered similar explanations regarding the relative importance of rainfall to runoff predictions. We also showed that the explanations are dependent on the machine learning models and the interpretation methods. The implication is that if we use these methods to investigate the involved hydrological processes, then various explanations are plausible. Schmidt et al. (2020) suggested the many possible explanations associated with different machine learning models are similar to the equifinality phenomenon in process-based hydrological modelling. The new results are discussed in greater detail in the updated manuscript.

[Figure]

Figure 1. Rainfall's contribution to runoff prediction at different time steps ahead. Each subfigure shows the results obtained in different outer cross-validation (CV) iterations and the mean values derived using a feature engineering method and an interpretation method.

Following the recent discussions on the "correct" methods to compute SHAP values in Chen et al. (2020) and Janzing et al. (2019), we used both the observational and the interventional methods to compute the SHAP values. The results obtained using the two methods, as shown in Figure 2, are overall similar. The implications and reasons to use each method are explained in the updated manuscript. The Python package "shap" was used in the computation, as it offers both methods (Lundberg et al., 2017). The source code for computation will be posted on GitHub when submitting the revision.

[Figure]

[Figure]

Figure 2. Rainfall's contribution to runoff prediction at different time steps ahead. Each subfigure shows the results obtained in different outer cross-validation (CV) iterations and the mean values derived using a feature engineering method and a SHAP computation method.

*(3) In light of comments #1 and #2, other hydrological studies using boosting or random forests while also emphasizing on interpretability (by using variable importance measures) could be discussed (in comparison to the present study) somewhere in the manuscript. What is the added value of the present work with respect to such existing works?*

In the updated manuscript we added a short review of the methods applying boosting and random forests methods in hydrology.

The main contribution of this study is as follow. (a) It presents the applications of model-agnostic local interpretation methods for interpreting individual predictions of rainfall-runoff models, whereas existing studies mostly use model-dependent global interpretation methods. (b) This study proposes a feature engineering and model training method to automatically find the optimal lower-dimensional representations of high-dimensional input time series for machine learning models. The contribution of the rainfall at each step to a runoff prediction can be easily computed using the proposed method. (c) This study shows that machine learning methods can be effective for modelling the rainfall-runoff correlations of SuDS. It also shows that the hydrological processes inferred by interpretation methods are dependent on the

machine learning models and the interpretation methods, i.e., there are many different but likely explanations to the predictions of the same event. Relevant discussions were added in the updated manuscript.

*(4) In the "Introduction" section, it is written that "only a few studies adopted machine learning methods to investigate the hydrological processes of SuDS", with the studies by Eric et al. (2015), Khan et al. (2013), Li et al. (2019), and Yang and Chui (2019) being discussed as examples of such studies. Since such studies are quite close to the present work, more of them could be reported (provided that they exist).*

Currently, there are very few studies applying machine learning methods to study the hydrological performances of sustainable urban drainage systems (SuDS). We explained the reasons behind the lack of popularity, and listed all the literature we can find in the updated manuscript, e.g., Hopkins et al. (2020) was added. More details of these studies are also presented in the updated manuscript. Additionally, we clarified our contributions more clearly (see our response to comment #3).

*(5) In the same section, it is also written that "modeling the responses of SuDS at fine temporal scales requires high-dimensional hydrometeorological time series to be used as input, which is difficult in machine learning". Could this sentence be further elaborated? I would say that the opposite holds, i.e., that machine learning methods are ideal for handling high-dimensional hydrometeorological time series.*

Our original statement was inaccurate. There are some machine learning methods that are very efficient in handling high dimensional data, such as deep learning methods. However, as discussed in Nielsen (2019), high-dimensional time series data are usually converted to lower-dimensional features before feeding to a machine learning model, unless the model is specifically designed to model sequence data. We were also not sure if the XGBoost method works well with high dimensional time series. Therefore, we designed a very flexible feature engineering method, that certain hyperparameter values would generate rainfall depth features that are very close to the original rainfall time series. The final features chosen were those corresponded to the highest prediction accuracy, and they were generally in low dimensions. In the updated methods, we updated this statement by saying that modelling high dimensional data can be challenging for some machine learning methods. And in the conclusion section, we suggested future studies to explore the usefulness of machine learning methods that are specifically designed for high dimensional sequence data, such as LSTM networks in deep learning.

*(6) The reader could also be referred to several specialized books (e.g., Hastie et al. 2009; James et al. 2013; Witten et al. 2007), for further information on the machine learning (or statistical learning) methods used in the paper.*

Thank you for your suggestion. In the updated manuscript, we provided a list of suggested references for the machine learning methods and the model interpretation methods.

*(7) Another concern of mine is related to the small number of real-world cases examined in the paper. I think that the application of the proposed procedures to large real-word datasets (comprising hundreds of cases) should be addressed at least with extensive relevant discussions in the manuscript (e.g., future research recommendations). (Currently, it is only suggested using "the SHAP method in more case studies"). To my view, these extensive discussions are important, especially given that (i) there are studies in the hydro-meteorological literature validating their models using big datasets, and (ii) the first aim of the paper is to "evaluate the usefulness of machine learning methods in predicting the hydrological responses of SuDS at fine temporal scales". The necessity of evaluating machine learning methods using big datasets is extensively discussed by Boulesteix et al. (2018).*

We agree that the proposed method should be thoroughly tested on different datasets to prove its usefulness. However, to the knowledge of the authors, there are no publicly available regional or global datasets on the rainfall and runoff time series of SuDS.

We have the rainfall-runoff data of SuDS for a few sites in the U.S., and the proposed methods were found to be effective for these sites. The two sites, WS and SHC, were chosen to be reported in the manuscript for the following reasons. (a) The two sites are in very different scales: WS is about 1,000 $m^2$, and SHC is about 1 $km^2$. We intended to show our methods are useful for catchments of various scales. (b) A few years of data were available for WS, and only two months of runoff data were available for SHC. We aimed to show our methods can be useful even when the data is not abundant. (c) The two sites faced different difficulties in setting up process-based models: the physical properties of the SuDS were unknown in WS, and the drainage system of SHC was very complex and the characterization of which requires thousands of parameters. We believe these difficulties are common in practice, and thus we presented the proposed methods as potential solutions to the common problems. The reasons and implications for choosing the two sites were clarified in the updated manuscript.

To address this comment, we suggest the proposed methods to be tested on more SuDS sites in the conclusion section of the paper. We also added discussions on usefulness of the proposed method in other fields of hydrology, where regional and global data, such as the CAMELS

dataset (Newman et al., 2015), are available. We also plan to include results of more SuDS cases studies as the demonstration applications in the documentation of the source code on GitHub.

*(8) In the "Conclusions" section it is written that "the proposed model training methods are semi-automatic, requiring minimal user input". It would be useful to discuss (somewhere in the paper) which parts of the proposed methods are not (fully) automatic, and how one could overcome this limitation to allow large-scale (even global-scale) investigations (see also comment #7).*

In response to this comment and the comments made by Anonymous Referee #1, we used Bayesian optimization algorithms to automatically find the optimal features and hyperparameters for training machine learning models (Snoek et al., 2012). This eliminates the need to select a predefined set of candidate feature engineering and XGBoost hyperparameters. The updated methods thus only require the lower and upper bounds for each hyperparameter. The user can also use the default values, if she/he so desires (the method then becomes fully automatic). This change allows the method to use regional scale data, where multiple sites were analysed. Relevant discussions are added in the updated manuscript.

The quality of the models derived using the updated method was found to be similar or better when comparing to that derived using the old methods (some of the results are presented in our responses to comment #1 made by Referee #1).

*(9) Currently, the use of the xgboost and SHAPforxgboost R packages is reported in the manuscript. To my view, all utilized software packages (which, of course, at the moment can be found online at https://github.com/stsfk/explainable_ml_hydro, since the R code has been made available) should necessarily be reported and cited in the paper.*

In response to this comment, we listed all the packages used in modelling in the updated manuscript.

In addition, we updated the source code using the "tidymodels" R packages (Kuhn and Silge, 2020), the source code is now easier to understand. We are also updating the documentation of the source code. The code and the documentation will be posted on GitHub upon submission of the revision.

*(10) Finally, the manuscript is not typo-free at the moment. Particular attention should be placed on the mathematical notations. For instance, the transpose operator should not be written in italics (therefore, T should be replaced with* T*) and the vectors should be bolded (therefore, Xt−m,t should be replaced with* **Xt−m,t***).*

Thank you for catching the errors. We will thoroughly check the manuscript when submitting the revised version and also hire a professional English editor to correct grammatical mistakes.

**Reference**

Chen, T., and Guestrin, C.: Xgboost: A scalable tree boosting system, Proceedings of the 22nd acm sigkdd international conference on knowledge discovery and data mining, 785-794, 2016

Chen, H., Janizek, J. D., Lundberg, S. and Lee, S. I.: True to the model or true to the data?, http://arxiv.org/abs/1805.11783, 2020.

Hopkins, K. G., Bhaskar, A. S., Woznicki, S. A. and Fanelli, R. M.: Changes in event-based streamflow magnitude and timing after suburban development with infiltration-based stormwater management, Hydrol. Process., 34(2), 387–403, https://doi.org/10.1002/hyp.13593, 2020.

Janzing, D., Minorics, L. and Blöbaum, P.: Feature relevance quantification in explainable ai: A causal problem, arXiv, 2019.

Kuhn, M. and Silge, J.: Tidy Modeling with R, Version 0.0.1.9007, https://www.tmwr.org, 2020.

Lundberg, S. and Lee, S.-I.: An unexpected unity among methods for interpreting model predictions, http://arxiv.org/abs/1611.07478, 2016.

Lundberg, S. M., Allen, P. G. and Lee, S.-I.: A Unified Approach to Interpreting Model Predictions, https://github.com/slundberg/shap, 2017.

Lundberg, S. M., Erion, G., Chen, H., DeGrave, A., Prutkin, J. M., Nair, B., Katz, R., Himmelfarb, J., Bansal, N. and Lee, S.-I.: From local explanations to global understanding with explainable AI for trees, Nat. Mach. Intell., 2(1), 56–67, https://doi.org/10.1038/s42256-019-0138-9, 2020.

Newman, A. J., Clark, M. P., Sampson, K., Wood, A., Hay, L. E., Bock, A., Viger, R. J., Blodgett, D., Brekke, L., Arnold, J. R., Hopson, T. and Duan, Q.: Development of a large-sample watershed-scale hydrometeorological data set for the contiguous USA: data set characteristics and assessment of regional variability in hydrologic model performance, Hydrol. Earth Syst. Sci., 19(1), 209–223, https://doi.org/10.5194/hess-19-209-2015, 2015.

Nielsen, A.: Practical Time Series Analysis, O'Reilly Media, Inc, https://www.oreilly.com/library/view/practical-time-series/9781492041641, 2019.

Ribeiro, M. T., Singh, S. and Guestrin, C.: Model-Agnostic Interpretability of Machine Learning, http://arxiv.org/abs/1606.05386, 2016.

Schmidt, L., Heße, F., Attinger, S. and Kumar, R.: Challenges in Applying Machine Learning Models for Hydrological Inference: A Case Study for Flooding Events Across Germany, Water Resour. Res., 56(5), https://doi.org/10.1029/2019WR025924, 2020.

Snoek, J., Larochelle, H. and Adams, R. P.: Practical Bayesian optimization of machine learning algorithms, in Advances in Neural Information Processing Systems, vol. 4, 2951–2959, https://arxiv.org/abs/1206.2944v2, 2012.

---

## Author Response (AR1)

**Authors' response to Anonymous Referee #1**

*Italicized text: comments made by Referee #1.*
Blue text: Authors' responses. The line numbers mentioned below correspond to those in the revised version of the manuscript.

*1. Firstly, this work is innovative for explaining machine learning predictions in hydrology forecasting. With applying AI in various fields and getting excellent results, it is a hot topic to interpret the machine learning. But this manuscript still has some questions needed revised. Generally, it is a good research point, but manuscript is hard to understand.*

The referee commented that our manuscript is hard to understand. To improve the readability of the paper, we updated the manuscript in the following aspects: (a) simplification of the methods, (b) removal of non-essential findings, and (c) re-organizing the paper according to the updated research objective. Almost the whole paper has been rewritten. The details are as follows.

(a) Simplification of the methods. We updated the feature engineering method and the hyperparameter optimization methods for training machine learning models.

In the revised manuscript, the high-resolution rainfall time series is converted into rainfall depth features using three hyperparameters, $m$, $l$, and $n$. Only the rainfalls recorded between $t - m$ and $t - 0$ are considered. Each rainfall depth recorded between $t - l$ and $t - 0$ is used for creating a rainfall depth feature. And $n$ intervals are created for aggregating the rainfall recorded between $t - l - 1$ and $t - m$, and the intervals roughly form an arithmetic sequence. See the illustration in Figure 1. The updated method is easier to understand, and the complex equations (Eq. 5 to Eq. 7) in the original submission were removed. More details of the updated feature engineering method are provided in Section 2.2.2, lines 214 to 236.

[Figure]

Figure 1. (a) Illustration of the methods to place cut points along the time axis. (b) Illustration of the four aggregation options for creating rainfall depth features after the cut points are selected.

We also simplified the hyperparameter optimization method. In particular, in the revised manuscript, we no longer differentiate the feature engineering hyperparameters and the XGBoost hyperparameters, and all the hyperparameters were optimized together through an automated Bayesian optimization method (Snoek et al., 2012). Figure 2 shows that the Bayesian optimization method can find high-quality solutions (as indicated by low inner cross-validation (CV) errors) in a few optimization steps.

[Figure]

Figure 2. (a) The model's prediction accuracy associated with the hyperparameters evaluated during each optimization step. The prediction accuracy is measured by the root-mean-square error (RMSE) of the predictions obtained during the inner cross-validation (CV) iterations. (b) The expected utility of the candidate hyperparameters evaluated during each optimization step.

We also investigated whether the optimization method resulted in overfitting the model selection criterion. As indicated by the positive correlation between the inner and outer CV errors in Figure 3 (i.e., the good models found during the optimization process also had good performances during testing), the optimization method did not overfit the model selection criterion. Readers do not need to understand the technical details regarding Bayesian optimization, and the method is described briefly in the revised manuscript (this point is mentioned in lines 334 to 344 in the updated manuscript). Due to the adoption of the updated methods, the descriptions on the resampling scheme, the choice of feature engineering and XGBoost hyperparameters, and the model selection methods were shortened. The models derived using the updated and the old methods were found to have comparable prediction accuracies to those found in the original submission.

[Figure]

Figure 3. The model's prediction errors estimated during the inner and the outer cross-validation (CV) iterations for each set of candidate hyperparameters evaluated during the optimization. The prediction errors are measured by root-mean-square error (RMSE). Each subfigure shows the result obtained for a rainfall depth feature aggregation method and during an outer CV iteration.

(b) Removal of non-essential findings. In the updated manuscript, the results of XGBoost hyperparameters optimization (Section 3.1 of the original submission) were removed due to the updated methods and the new research objectives. The results on interpreting the feature engineering hyperparameters (Section 3.3.1 of the original submission) were also removed due to their indirect connections to the hydrological processes. The descriptions of goodness-of-fit of the trained models (Section 3.2 of the original submission) were shortened.

(c) Re-organizing the paper according to the new research objective. We removed the following research objective from the updated manuscript, "develop and present tools and methods for building higher quality machine learning models for SuDS-related studies and demonstrate the applications". And the new objective is "to evaluate the usefulness of explainable machine learning methods in modeling and interpreting the hydrological responses of SuDS to rainfall at sub-hourly time scales". Therefore, we removed the "demonstration" element from the original submission and focused on developing methods for modelling SuDS and interpreting the model predictions and presenting findings regarding the accuracy and the interpretability

of the machine learning models. The following changes were made in terms of the content of the paper.

In the introduction section of the updated manuscript, an introduction to the applications of interpretation methods in hydrology is presented (lines 94 to 109).

In the methods section, we extended the introduction to methods for interpreting machine learning models. The differences between the local and the global interpretation methods (such as the commonly used gain and cover metrics for XGBoost) were first discussed, and the two methods (the observational and the conditional expectation methods) to compute the SHAP values and their implications were introduced. The methods to assign the importance and the contribution scores of the rainfall of each time step to runoff prediction were then introduced. A method to combine the explanations of multiple input samples was subsequently introduced. The other commonly used global interpretation methods were then presented. The descriptions of machine learning model training methods, including nested-cross validation procedure and hyperparameter optimization methods, were substantially shortened. More focus was given to the interpretation methods, as literature on this topic is currently lacking in the field of hydrology, and readers can get a better understanding of the results if more information on the interpretation methods is provided.

In the results section, we removed some of the results on machine learning model training and the results on interpreting feature engineering hyperparameters. Results on the prediction accuracies of the models were first presented (Section 3.1). Regarding the application of the interpretation methods, the results on interpreting the global model structures were first presented, where a comparison between SHAP and other global interpretation methods was also added, as suggested by Referee #2 (Section 3.2). This results on examining the models' structures for predicting discharges of different magnitudes is then presented (Section 3.3). Finally, a few applications using the local explanations (i.e., SHAP values for individual input samples) were presented, including hydrograph decomposition, and determination of the source of predicted discharge (Section 3.4). Section 3.2 through Section 3.4 were organized according to the aggregation levels of the explanations, from global levels (all samples) to multiple samples levels, then to region levels (each sample).

*2. The logic of this paper is not clear that I cannot figure out what information explained by SHAP model and what relationship of hydrological response and selected hyperparameters.*

The SHAP method is a feature attribution method, i.e., it aims to explain how much each feature contributed to the output of a model for a particular sample (Janzing et al., 2019). For this study, the contribution of rainfall of each time step to each discharge prediction is

computed. In the updated manuscript, we added a formal introduction to the feature attribution problem and their methods (Section 2.1.2). In Section 3.2, when presenting the comparison between the results produced by the observational and the interventional SHAP, we commented that,

"The contributions resulting from the observational SHAP values can be interpreted as the expected difference in the predicted discharge when a particular rainfall is observed/not observed, accounting for the presence/absence of the other rainfall measurements." (lines 547 to 549)

"The contributions that result from the interventional SHAP values are the expected prediction changes when rainfall is set to a specific value, accounting for the presence/absence of other rainfall measurements." (lines 550 to 552)

In response to this particular comment, we removed the content on explaining the hyperparameters in the updated manuscript, as we found their connections to the hydrological processes are indirect. The updated manuscript only explains the contribution of rainfalls of each time step to each runoff prediction.

*3. I think the main question is limited input variables (only Rainfall depth). I cannot agree that the design rainfall depth features (Section 2.1.1) reflect SuDS hydrological process. Thus, the hyperparameters of m, l, q, account_CumRain and account_season have little meaning for interpreting hydrological process in SuDS. Originally, SHAP is a game theoretic approach to explain the output of machine learning model. So maybe more physical observation variables are needed to selected as input variables. Therefore, I suggest this manuscript for Major Revision and Resubmission.*

We agree with this comment that more variables can be included in the machine learning models. However, for the study site, only the rainfall and the runoff time series were available. The lack of data (such as the physical properties of the catchment) for setting up process-based models was also a motivation for using machine learning methods. In addition, this study focuses on modelling stormwater runoffs of small-scale urban drainage infrastructures during the wet period (i.e., within 24 hours of rainfall events), thus rainfall is the main driver of the system being modelled. Studying how a runoff prediction is affected by the rainfall of each time step is meaningful. Moreover, we also considered additional input features to the machine learning models to account for the potential seasonality of the performance of the SuDS. Finally, this study recommends future studies to include more variables in machine learning models in the conclusion section.

In response to this comment, we removed the content on explaining the hyperparameters, as they are indirectly connected to the hydrological processes of SuDS.

In the updated manuscript, we presented SHAP as a method to explain the basis of a prediction for checking whether that prediction can be trusted. That is, we do not think the hydrological processes inferred by machine learning models are necessarily true. We further clarified this point by adding studies comparing the inferred hydrological processes of different machine learning models derived using different methods for computing SHAP values. As an example, Figure 4 shows that different machine learning models considered the rainfall's contributions to runoff predictions differently. Thus, there were considerable uncertainties in interpreting machine learning model predictions. The existence of various possible explanations was referred to as equifinality in Schmidt et al. (2020), which is an important concept in hydrological modelling (Beven and Freer, 2001). We reported this issue in the updated manuscript in lines 542 to 545.

[Figure]

Figure 4. The average (a) importance and (b) contribution of the rainfall of different time step in the past for discharge predictions in different models. Each subfigure shows the results obtained in different outer cross-validation (CV) iterations and the mean values derived using a feature engineering method and a SHAP computation method.

In this study, we showed that the interpretation method can be useful tools for model diagnosis, for instance, the models built for SHC were found to be inadequate, as the weights they assigned to the rainfall depths at each time step during prediction were not realistic (lines 562

to 565). The interpretation methods were also used to solve some hydrological tasks, such as hydrograph decomposition. We do not consider the hydrological processes inferred using the interpretation methods are true, we commented on their shortcomings,

"However, this hydrograph decomposition method has some limitations. First, negative contributions are assigned to the rainfalls, which are difficult to interpret. Second, there is a constant bias term in the hydrograph, which does not correspond to any element in the commonly used conceptual rainfall–runoff models. Third, a model might use features that are not derived from rainfall (e.g., temperature) as a predictor, which will also be assigned with contributions to the runoff predictions when SHAP methods are used. It is unclear how to represent the contributions of factors other than rainfall when decomposing a hydrograph. Nevertheless, the hydrograph decomposition results shown in Fig. 9 are still useful for understanding why a given prediction is made by the model and to what extent the rainfall of each time step contributes to the model runoff prediction." (lines 638 to 645)

*4. Point 1:Whether the constructed data feature mining algorithm corresponds to the reference standard in the folded data part?*

In the updated manuscript, we no longer discuss the specific values of feature engineering hyperparameters due to their indirect connections to the hydrological processes. We used a Bayesian optimization method to optimize the hyperparameters automatically. The updated optimization methods are described in lines 334 to 344.

*5. Point 2: "The framework is particularly useful for urban catchments where the information for setting up process-based models is insufficient." Is this statement reasonable? Do similar expressions still exist in the full text?*

Thank you for raising this question. We think further clarification is needed in the updated manuscript. We claimed that,

"It was difficult to build process-based hydrological models for the two SuDS catchments examined in this study due to insufficient information regarding the physical properties and drainage processes. This study designs a simple feature engineering method to facilitate the application of the commonly used machine learning algorithms to model rainfall–runoff correlations at sub-hourly time scales". (lines 688 to 691)

As for the "insufficient information", we mean that the physical properties of the drainage systems in the first case study were unknown for the first site, and it was also difficult to

represent the unknown leakage from SuDS using process-based models. The second case study site contains a large-scale and complex drainage network, requiring many parameters to characterize their physical properties in process-based models. We mention these points in the updated manuscript in lines 389 to 401. A table was added to the updated manuscript to compare the characteristics of the selected study sites (Table 1). Machine learning models were built relatively easily in the two case studies and showed relatively high prediction accuracies. However, we argue that process-based hydrological models are useful for examining the involved hydrological processes. We pointed out the shortcomings and the uncertainties related to the data-driven approach (see our responses to comment #4).

Therefore, machine learning methods are useful for modelling the statistical correlations between interested random variables of the catchment when observation data of the variables are available, and the trained machine learning models can serve as a baseline for evaluating process-based models. We removed the original statement in the updated manuscript.

*6. Point 3: Adding quantitative analysis to the conclusion section should be more convincing.*

In the updated manuscript, we added the performance metrics of the trained machine learning models in the abstract and the conclusion section.

"The resulting models have high prediction accuracies (the Nash–Sutcliffe model efficiency coefficient (NSE) > 0.70 and the coefficient of determination ($R^2$) > 0.70 for all models)." (lines 17 to 18)

"The proposed methods are applied to two SuDS catchments of different sizes, SuDS practice types, and data availabilities to predict discharge and produce models with good prediction accuracies ($NSE > 0.7$ and $R^2 > 0.7$ for all models)." (lines 670 to 672)

*7. Point 4: Compared with the commonly used urban rainfall runoff models, what are the obvious advantages of this model?*

The advantage of using machine learning methods is that they only require observation data of the interested random variables and do not require the involved physical processes to be characterized. Machine learning models can generally be set up easily and can potentially provide high-quality predictions. Machine learning models may be used as baseline models for the evaluation of process-based models. More explanations are provided in our response to comment #5. In the Section 2.3.1, we discussed why machine learning models are used,

"Process-based models can be difficult to develop for both sites. In WS, the physical properties and exact design of the different drainage system elements are not precisely known (Darner et al., 2015). The rain garden is also not isolated from the gravel storage layer of the porous pavements, which permits an unknown amount of stormwater from the rain garden into the underdrain system of the porous pavement. However, commonly used process-based models are mostly designed to model SuDS with standard designs and may not be directly applicable to WS. In the SHC, the main challenge lies in the heavy workload and uncertainties in estimating the model parameters that characterize the complex drainage system. The SHC can be divided into multiple subcatchments connected by the drainage network, and a number of parameters must be determined for each catchment. In particular, the portions of impervious area that are directly and indirectly connected to the drainage network must be specified to accurately represent the flow paths of each subcatchment." (lines 389 to 397)

*8. Line 620-780: It is difficult for finding the references because of improperly format.*

Thank you for catching the errors. We updated the citation styles throughout the manuscript to meet HESS requirement.

*9. Line 9: How do you define the "fine temporal scales"? It is an important concept in your forecasting, but it is not clear.*

"Fine temporal scales" refers to "sub-hourly time scales". In the updated manuscript, the term "sub-hourly time scales" was used as it is more specific.

*10. Line 131: Why you use Dt-a,t-b for aggregating rainfall depth?*

The lower-dimensional rainfall depth features $D_{t-a,t-b}$ were used because the dimension of the original rainfall time series can be very high (e.g., 1,000 time steps), and some machine learning methods have difficulties to learn the high-dimensional correlation between the input and the output random variables. Thus, we designed a feature engineering method to lower the dimension of the input variables of the machine learning models, and the number of features used is controlled by three hyperparameters. In fact, the feature engineering method allows the rainfall depth features to be very similar to the original time series. And the values of the hyperparameters are chosen according to the prediction accuracy of the resulted machine learning models. This point is explained in the updated manuscript.

"The hydrological responses of SuDS can be affected by relatively long-term hydrometeorological conditions that occurred in the past. $X_t$ can thus be a long time series of hydrometeorological condition measurements. As pointed out by Nielsen (2019), many machine learning algorithms are not designed for modeling time series data. Therefore, long time series are often converted into lower-dimensional features that are then used as the input variables of machine learning models. The input variable transformation process is known as feature engineering and is expected to produce higher-quality models (Kuhn and Johnson, 2019)." (lines 197 to 202)

"Representing a rainfall time series using a set of $D_{t-a,t-b}$ can reduce the data dimension at the cost of losing information regarding the temporal rainfall distribution. Note that fewer cut points are selected for rainfalls in the long-term past (e.g., a few days), implying that they play less important roles in predicting $Y_t$. This is reasonable considering the relatively fast response time of SuDS (DeBusk et al., 2011). Gauch et al. (2020) also showed that the hydrometeorological time series recorded in the long-term past can be represented using a coarser temporal resolution in machine learning models built for rainfall–runoff modeling without deteriorating their prediction accuracy. In the proposed method, the three hyperparameters and aggregation options control the aggregation level and approach by which rainfall data recorded at different time steps are aggregated." (lines 229 to 236)

*10. In Line 84 said many observation data became available, but why only the rainfall data? Do you have other data?*

We only have rainfall and runoff data for both study sites, as reported in our response to comment #3. The sentence "more observation data became available" was referring to the fact that the rainfall and runoff are being monitored in more SuDS sites globally. However, as pointed out by Schaffitel et al. (2020), monitoring data of other variables concerning urban hydrology are still currently lacking. Therefore, it can be useful to present a study that focuses only on the correlation between rainfall and runoff time series. More information on this issue is also presented in our response to comment #3. We commented on this issue in the updated manuscript.

"There is a limited amount of publicly available data concerning the hydrological processes of SuDS because they represent relatively new technologies and monitoring is often conducted by the local authorities and other interested parties. A lack of data is also common for other data types that are useful in urban hydrological studies, such as the soil moisture content and soil temperature (Schaffitel et al., 2020). It may be therefore useful to demonstrate the applications of machine learning methods to predict SuDS discharge based on preceding rainfalls because rainfall and discharge are the most commonly monitored features in SuDS

sites and several rainfall–discharge datasets are available online (e.g., the United States Geological Survey Water Data for the Nation, https://waterdata.usgs.gov)." (lines 110 to 116)

*11. Line 6-14 and Line 560-595: In the section of abstract and conclusion, the quantitative results are absent and the qualitative descriptions are not enough.*

Quantitative results are presented in the updated manuscript. See our response to comment #6.

**Reference**

Beven, K. and Freer, J.: Equifinality, data assimilation, and uncertainty estimation in mechanistic modelling of complex environmental systems using the GLUE methodology, J. Hydrol., 249(1–4), 11–29, https://doi.org/10.1016/S0022-1694(01)00421-8, 2001.

Janzing, D., Minorics, L. and Blöbaum, P.: Feature relevance quantification in explainable AI: A causal problem, https://arxiv.org/abs/1910.13413, 2019.

Schaffitel, A., Schuetz, T. and Weiler, M.: A distributed soil moisture, temperature and infiltrometer dataset for permeable pavements and green spaces, Earth Syst. Sci. Data, 12(1), 501–517, https://doi.org/10.5194/essd-12-501-2020, 2020.

Schmidt, L., Heße, F., Attinger, S. and Kumar, R.: Challenges in Applying Machine Learning Models for Hydrological Inference: A Case Study for Flooding Events Across Germany, Water Resour. Res., 56(5), https://doi.org/10.1029/2019WR025924, 2020.

Snoek, J., Larochelle, H. and Adams, R. P.: Practical Bayesian optimization of machine learning algorithms, in Advances in Neural Information Processing Systems, vol. 4, pp. 2951–2959, https://arxiv.org/abs/1206.2944v2, 2012.

**Authors' response to Referee #2–Dr. Georgia A Papacharalampous**

*Italicized text: comments made by Dr. Georgia A Papacharalampous (Referee #2).*
Blue text: Authors' responses. The line numbers mentioned below correspond to those in the revised version of the manuscript.

***Summary****: The paper focuses on the predictive modeling of sustainable drainage systems (SuDS) at fine temporal scales using boosting (Friedman 2001). Several boosting variants are formed and exploited in two case studies, while comparisons with the linear regression algorithm and the Storm Water Management Model (SWMM; Rossman 2015) are also provided. Furthermore, the SHapley Additive exPlanations (SHAP) method (Lundberg and Lee 2017) is used to explain the contribution of each variable (else referred to as "feature") to the issued predictions, thereby facilitating interpretability to some extent.*

Thank you for providing a nice summary of our research.

***General comments:*** *In general, I find that the manuscript is well-formulated and -written, and I think that the work done so far (including the release of the R codes at GitHub) should be appreciated. Nonetheless, I also think that there is some room for improvement before publication.*

*I recommend minor revisions. My comments are given right below.*

Thank you very much for the positive assessment. We have revised the manuscripts according to your comments. In particular, we added an introduction to machine learning model interpretation methods in the introduction sections (lines 94 to 109), which can help the readers understand the findings of this study better. We also added a comparison between the proposed SHAP-based interpretation method and the commonly used feature importance assessment methods (lines 279 to 295, and lines 575 to 594). The source code provided on GitHub has also been updated, where more explanations were provided. The proposed methods have been tested on several additional case studies, the source code and the results can be found on GitHub.

In addition, in response to the comments raised by Anonymous Referee #1, we have updated the methods and substantially changed the paper structure. (a) The feature engineering and hyperparameter optimization methods have been simplified. (b) The non-essential findings on the model training processes have been removed. (c) The structure of the paper has been reorganized, focusing more on explaining machine learning models. These modifications did not change the overall content and the conclusion of the paper. Detailed information can be found in Authors' response to Referee #1.

***Comments:***

*(1) To my view, the following clarification is required: Which are the similarities and differences between basic variable importance measures (available in the xgboost R package) and the SHAP methodology (available in the SHAPforxgboost R package)?*

In the updated manuscript, we added an introduction to the methods for interpreting machine learning models in Section 2.1. We showed that SHAP is a local feature attribution method, and the other commonly used feature importance measures (such as gain, cover, and frequency) are global interpretation methods. In addition, we showed that local explanations derived using SHAP can be combined to understand the global structures of the model in Section 2.2.3 to Section 2.2.4. And in Section 2.2.5, the other global feature importance measures are introduced.

The main differences between SHAP and the other feature importance measures are as follow:

(a) SHAP is a model-agnostic interpretation method, and the other importance measures provided by the "xgboost::xgb.importance" function in R package are model-specific (Chen et al., 2020). The advantage of model-agnostic methods is that they can be applied to various machine learning models and thereby allow comparisons between different types of models in terms of the derived interpretations (Ribeiro et al., 2016).

(b) SHAP is a local interpretation method while the other methods provided by the "xgboost::xgb.importance" function are global interpretation methods. The local interpretation methods are designed for interpreting the prediction made for individual input samples, and the global methods are independent of the input samples and often explain the structure of the model (Lundberg and Lee, 2016). Therefore, in this study, SHAP can be used to analyse a specific runoff prediction, and the other methods cannot be used for this task.

(c) Theoretically, SHAP is the only method that provides interpretations that satisfy a series of desired properties, such as local accuracy, missingness, and consistency (Lundberg et al., 2020).

*(2) Since interpretability is one of the main themes of the present work, I feel that a comparison (direct or indirect, depending on the answer to comment #1) between basic variable importance measures and the SHAP methodology is currently missing from the manuscript and should be necessarily made for both case studies. New computations are needed for this comment to be fully addressed (independently of the answer to comment #1); however, these computations will only require the xgboost R package (which is already used in the paper).*

Thank you for your suggestion. In the updated manuscript, we added a comparison between the feature importance derived using different interpretation methods (lines 575 to 589 and Figure 7).

The original submission, however, did not present the computation steps for deriving global feature importance measures from local explanations computed using SHAP. In the updated manuscript, we formally defined the methods to compute the contribution of rainfall at each time step to runoff predictions (Section 2.2.3, lines 238 to 254) and the method to combine local explanations to understand model structures and rainfall-runoff correlations (Section 2.2.4, lines 262 to 278). We believe the addition of the formal definitions can improve the readability of the paper.

In Figure 1, the importance of the rainfall of each time to discharge prediction obtained using various feature importance measures are compared. The results obtained using various methods are generally similar, confirming the validity of the opposed SHAP-based method. More discussions on this result can be found in lines 575 to 589.

[Figure]

Figure 1. The importance of rainfall of each time to runoff prediction. Each subfigure shows the results obtained in different outer cross-validation (CV) iterations and the mean values derived using a feature engineering method and an interpretation method.

Following the recent discussions on the "correct" methods to compute SHAP values in Chen et al. (2020) and Janzing et al. (2019), we used both the observational and the interventional methods to compute the SHAP values. The results obtained using the two methods, as shown in Figure 2, are overall similar. The implications and reasons to use each method are explained in the updated manuscript. The Python package "shap" was used in the computation, as it offers both methods (Lundberg et al., 2017). The two methods are introduced in lines 150 to 158, and more descriptions of the results can be found in lines 546 to 556 of the updated manuscript.

[Figure]

Figure 2. The average (a) importance and (b) contribution of the rainfall of different time step in the past for discharge predictions in different model. Each subfigure shows the results obtained in different outer cross-validation (CV) iterations and the mean values derived using a feature engineering method and a SHAP computation method.

*(3) In light of comments #1 and #2, other hydrological studies using boosting or random forests while also emphasizing on interpretability (by using variable importance measures) could be discussed (in comparison to the present study) somewhere in the manuscript. What is the added value of the present work with respect to such existing works?*

The main contribution of this study is as follows. (a) It presents the applications of model-agnostic local interpretation methods for interpreting individual predictions of rainfall-runoff models, whereas existing studies mostly use model-dependent global interpretation methods. (b) This study proposes a feature engineering and model training method to automatically find

the optimal lower-dimensional representations of high-dimensional input time series for machine learning models. (c) This study shows that machine learning methods can be effective for modelling the rainfall-runoff correlations of SuDS. (d) It defines various methods to use the local explanations derived by the SHAP method for model diagnosis and analysing the hydrological processes being modelled. We discussed the contributions of this paper in the conclusion section, lines 688 to 707.

*(4) In the "Introduction" section, it is written that "only a few studies adopted machine learning methods to investigate the hydrological processes of SuDS", with the studies by Eric et al. (2015), Khan et al. (2013), Li et al. (2019), and Yang and Chui (2019) being discussed as examples of such studies. Since such studies are quite close to the present work, more of them could be reported (provided that they exist).*

Currently, there are very few studies applying machine learning methods to study the hydrological performances of sustainable urban drainage systems (SuDS). We explained the reasons behind the lack of popularity (lines 72 to 116), and listed all the literature we can find in the updated manuscript, e.g., Hopkins et al. (2020) was added. More details of these studies are also presented in the updated manuscript. Additionally, we clarified our contributions more clearly (see our response to comment #3).

*(5) In the same section, it is also written that "modeling the responses of SuDS at fine temporal scales requires high-dimensional hydrometeorological time series to be used as input, which is difficult in machine learning". Could this sentence be further elaborated? I would say that the opposite holds, i.e., that machine learning methods are ideal for handling high-dimensional hydrometeorological time series.*

Our original statement was inaccurate. There are some machine learning methods that are very efficient in handling high-dimensional data, such as deep learning methods. However, as discussed in Nielsen (2019), high-dimensional time series data are usually converted to lower-dimensional features before feeding to a machine learning model, unless the model is specifically designed to model sequence data. We were also not sure if the XGBoost method works well with high-dimensional time series. Therefore, we designed a very flexible feature engineering method, that certain hyperparameter values would generate rainfall depth features that are very close to the original rainfall time series. The final features chosen were those that corresponded to the highest prediction accuracy, and they were generally in low dimensions. In the updated manuscript, we updated this statement by saying that modelling high dimensional data can be challenging for some machine learning methods.

"Thus, modeling the responses of SuDS at fine temporal scales requires a high-dimensional hydrometeorological time series to be used as input, which is difficult for machine learning algorithms that are not specifically designed for modeling data sequences (Nielsen, 2019)." (lines 88 to 91)

And in the conclusion section, we suggested future studies to explore the usefulness of machine learning methods that are specifically designed for high dimensional sequence data, such as LSTM networks in deep learning.

"However, feature engineering methods can be designed arbitrarily and it can be computationally expensive to identify the optimal methods. Future studies can thus explore the application of machine learning methods that are specifically designed for modeling high-dimensional time series data, such as the long short-term memory (LSTM) networks in deep learning." (lines 709 to 712)

*(6) The reader could also be referred to several specialized books (e.g., Hastie et al. 2009; James et al. 2013; Witten et al. 2007), for further information on the machine learning (or statistical learning) methods used in the paper.*

Thank you for your suggestion. In the updated manuscript, we provided a list of suggested references for the machine learning methods and the model interpretation methods. For instance,

"More information on the various interpretation methods can be found in Molnar (2021)." (line 132)

"More information on the Shapley value can be found in Osborne and Rubinstein (1994)." (lines 145 to 146)

"More information on the resampling techniques for testing machine learning models can be found in Kuhn and Johnson (2013) and Hastie et al. (2009)." (lines 354 to 355)

*(7) Another concern of mine is related to the small number of real-world cases examined in the paper. I think that the application of the proposed procedures to large real-word datasets (comprising hundreds of cases) should be addressed at least with extensive relevant discussions in the manuscript (e.g., future research recommendations). (Currently, it is only suggested using "the SHAP method in more case studies"). To my view, these extensive discussions are important, especially given that (i) there are studies in the hydro-meteorological literature validating their models using big datasets, and (ii) the first aim of*

*the paper is to "evaluate the usefulness of machine learning methods in predicting the hydrological responses of SuDS at fine temporal scales". The necessity of evaluating machine learning methods using big datasets is extensively discussed by Boulesteix et al. (2018).*

We agree that the proposed method should be thoroughly tested on different datasets to prove its usefulness. However, to the knowledge of the authors, there are no publicly available regional or global datasets on the rainfall and runoff time series of SuDS.

We have the rainfall-runoff data of SuDS for a few sites in the U.S., and the proposed methods were found to be effective for these sites. The two sites, WS and SHC, were chosen to be reported in the manuscript for the following reasons. (a) The two sites are in very different scales: WS is about 1,000 $m^2$, and SHC is about 1 $km^2$. We intended to show our methods are useful for catchments of various scales. (b) A few years of data were available for WS, and only two months of runoff data were available for SHC. We aimed to show our methods can be useful even when the data is not abundant. (c) The two sites faced different difficulties in setting up process-based models: the physical properties of the SuDS were unknown in WS, and the drainage system of SHC was very complex and the characterization of which requires thousands of parameters. We believe these difficulties are common in practice, and thus we presented the proposed methods as potential solutions to the common problems. The reasons and implications for choosing the two sites were listed in Table 1 in the updated manuscript.

To address this comment, we suggest the proposed methods be tested on more SuDS sites in the conclusion section of the paper.

"Second, the methods proposed in this study are only applied to model the correlations between rainfall time series and discharge in a few U.S. sites. The proposed methods should therefore be tested in more sites worldwide to model the correlations between other variables, although this may require the development of new feature engineering methods." (lines 712 to 715)

Finally, we added results of more SuDS cases studies as the demonstration applications in the documentation of the source code on GitHub.

*(8) In the "Conclusions" section it is written that "the proposed model training methods are semi-automatic, requiring minimal user input". It would be useful to discuss (somewhere in the paper) which parts of the proposed methods are not (fully) automatic, and how one could overcome this limitation to allow large-scale (even global-scale) investigations (see also comment #7).*

In response to this comment and the comments made by Anonymous Referee #1, we used Bayesian optimization algorithms to automatically find the optimal features and hyperparameters for training machine learning models (Snoek et al., 2012). This eliminates the need to select a predefined set of candidate feature engineering and XGBoost hyperparameters. The updated methods thus only require the lower and upper bounds for each hyperparameter. The user can also use the default values, if she/he so desires (the method then becomes fully automatic). This change allows the method to use regional scale data, where multiple sites were analyse,

"The model training process is automatic and only requires that the range of possible hyperparameter values be defined." (line 696)

The quality of the models derived using the updated method was found to be similar or better when comparing to that derived using the old methods (some of the results are presented in our responses to comment #1 made by Referee #1).

*(9) Currently, the use of the xgboost and SHAPforxgboost R packages is reported in the manuscript. To my view, all utilized software packages (which, of course, at the moment can be found online at https://github.com/stsfk/explainable_ml_hydro, since the R code has been made available) should necessarily be reported and cited in the paper.*

In response to this comment, we listed the packages used in this paper in the code availability section. In addition, we updated the source code using the "tidymodels" R packages (Kuhn and Silge, 2020), the source code is now easier to understand.

*(10) Finally, the manuscript is not typo-free at the moment. Particular attention should be placed on the mathematical notations. For instance, the transpose operator should not be written in italics (therefore, T should be replaced with* T*) and the vectors should be bolded (therefore, Xt−m,t should be replaced with **Xt−m,t**).*

Thank you for catching the errors, we updated the math notations accordingly. The manuscript has been thoroughly checked. We also hired a professional English editor to correct grammar mistakes.

**Reference**

Chen, T., and Guestrin, C.: Xgboost: A scalable tree boosting system, Proceedings of the 22nd acm sigkdd international conference on knowledge discovery and data mining, 785-794, 2016

Chen, H., Janizek, J. D., Lundberg, S. and Lee, S. I.: True to the model or true to the data?, http://arxiv.org/abs/1805.11783, 2020.

Hopkins, K. G., Bhaskar, A. S., Woznicki, S. A. and Fanelli, R. M.: Changes in event-based streamflow magnitude and timing after suburban development with infiltration-based stormwater management, Hydrol. Process., 34(2), 387–403, https://doi.org/10.1002/hyp.13593, 2020.

Janzing, D., Minorics, L. and Blöbaum, P.: Feature relevance quantification in explainable ai: A causal problem, arXiv, 2019.

Kuhn, M. and Silge, J.: Tidy Modeling with R, Version 0.0.1.9007, https://www.tmwr.org, 2020.

Lundberg, S. and Lee, S.-I.: An unexpected unity among methods for interpreting model predictions, http://arxiv.org/abs/1611.07478, 2016.

Lundberg, S. M., Allen, P. G. and Lee, S.-I.: A Unified Approach to Interpreting Model Predictions, https://github.com/slundberg/shap, 2017.

Lundberg, S. M., Erion, G., Chen, H., DeGrave, A., Prutkin, J. M., Nair, B., Katz, R., Himmelfarb, J., Bansal, N. and Lee, S.-I.: From local explanations to global understanding with explainable AI for trees, Nat. Mach. Intell., 2(1), 56–67, https://doi.org/10.1038/s42256-019-0138-9, 2020.

Nielsen, A.: Practical Time Series Analysis, O'Reilly Media, Inc, https://www.oreilly.com/library/view/practical-time-series/9781492041641, 2019.

Ribeiro, M. T., Singh, S. and Guestrin, C.: Model-Agnostic Interpretability of Machine Learning, http://arxiv.org/abs/1606.05386, 2016.

Schmidt, L., Heße, F., Attinger, S. and Kumar, R.: Challenges in Applying Machine Learning Models for Hydrological Inference: A Case Study for Flooding Events Across Germany, Water Resour. Res., 56(5), https://doi.org/10.1029/2019WR025924, 2020.

Snoek, J., Larochelle, H. and Adams, R. P.: Practical Bayesian optimization of machine learning algorithms, in Advances in Neural Information Processing Systems, vol. 4, 2951–2959, https://arxiv.org/abs/1206.2944v2, 2012.

---

## Referee Report (RR1)

**Review of Modeling and interpreting hydrological responses of sustainable urban drainage systems with explainable machine learning methods.**

The article by Yang and Chui shows the results of a study on the prediction of the hydrological response of sustainable urban drainage systems (SuDS) using Machine Learning algorithms.

Through an in-depth examination of both the manuscript and the authors 'responses to other reviewers' comments, I was able to appreciate the effort made by the authors to adequately address the constructive comments of colleagues.

However, I believe that the article is not yet ready for publication and should be re-evaluated after further review.

I expect the Authors to further improve the article in order to overcome my main concerns:

- First, I believe that a Machine Learning-based approach is less suitable for addressing a SuDS problem than a physically based systemic approach. If it is true that in some cases the geometric and hydraulic characteristics of these systems are not well known, it is even more true that the presence of experimental field data on inflows and outflows represents a rare exception. For this reason, a physically based modeling is in most cases to be preferred and allows to address a wider variety of problems, while in this case a model based on Machine Learning algorithms would be limited to the specific case. Authors should give more convincing reasons for the choice of approach.

- The novelty of the work is not relevant from a methodological point of view. The XGBoost algorithm is widely used in literature, as well as the other tools (Nested cross-validation, Bayesian optimization, etc.) used in modeling. Authors should better highlight in the introduction section why this work would represent a significant upgrade over existing literature, worthy of publication in HESS.

- The article is very long, and in some places still unclear. It could certainly be shortened without compromising its contents. This is the most important aspect on which Authors should focus their efforts in order to obtain a more concise and clear manuscript.

Furthermore, I would ask the Authors to consider the following additional comments:

P3 L67 - "Machine learning methods, also referred to as data-driven modeling, predictive modeling, and statistical learning": These terms are not strictly equivalent.

P4 L126 – Section 2.1.1 Local and global methods:  This section is very basic and not essential for the manuscript. Sections 2.1.1 and 2.1.2 should be merged and shortened.

P14 L183 – "The water levels are converted into discharge measurements using stage-discharge rating curves": How were the curves obtained? Do they come from an appropriate calibration?

P16 L426 – "The feature engineering and XGBoost hyperparameters are automatically optimized using the Bayesian optimization": Authors should report the optimal values of the hyperparameters, possibly in a table.

P18 L486 – Nash-Sutcliffe efficiency coefficient and coefficient of determination $R^2$ are very similar metrics, it is not useful to consider both.

P28 L668 – Conclusions: This section should be much more concise and effective in summarizing the main findings of the study.

---

## Author Response (AR2)

**Authors' response to Anonymous Referee #3**

*Italicized text: comments made by Referee #3.*
Blue text: Authors' responses. The line numbers mentioned below correspond to those in the revised version of the manuscript.

*The article by Yang and Chui shows the results of a study on the prediction of the hydrological response of sustainable urban drainage systems (SuDS) using Machine Learning algorithms.*

*Through an in-depth examination of both the manuscript and the authors 'responses to other reviewers' comments, I was able to appreciate the effort made by the authors to adequately address the constructive comments of colleagues.*

*However, I believe that the article is not yet ready for publication and should be re-evaluated after further review.*

Thank you very much for reviewing our paper and providing insightful suggestions. We have revised our paper carefully according to your comments and the comments from the editor. The main changes to the manuscript are as follows.

1. Improved presentation. (a) The paper is shortened by around 25%, i.e., about 3,000 words have been removed. (b) We reduced the use of equations and jargon from the machine learning (ML) field. 5 equations have been removed. (c) We removed the detailed introduction to explanation methods, which are too basic and not essential for understanding the study results. (d) We added three new figures (Figures 1, 3, and 10) to graphically illustrate the model training processes, the numerical experiments to be conducted, and the processes of inferring the physical processes being modeled from ML modeling results. (e) The figures are easier to read due to the removal of redundant results.

2. More in-depth discussions on the practices of inferring the physical processes being modeled from ML modeling results. In section 3.6, we presented a high-level overview of the practices of making inferences based on the model explanations. We show that there are large uncertainties involved in every step of the inference process, which are often overlooked in current hydrological ML studies. We consider this to be an important message of this paper.

3. Removal of non-essential findings. (a) In the previous submission, four feature engineering methods were used; however, they are overall similar. In the updated manuscript, only one method is used. (b) We removed the content on testing whether overfitting occurs in the optimization process. (c) The content on the estimation of "water age" has been remove as they may be considered redundant to the catchment response time estimation results.

4. Better organization of the results. In the previous submission, we did not raise an interesting research question that connects different parts of the paper, so that the results seem to be random. In the updated manuscript, we shift our focus from building more accurate models to testing whether the explanations derived from ML modeling results are consistent with our hydrological knowledge. The four numerical experiments to be conducted are described

clearly in the method section, and all of them are closely related to the main questions proposed in the introduction section.

5. New numerical experiments on analyzing the correlation between a model's prediction accuracy and the physical realism of the inferred physical processes. This experiment is added because we intended to show that an ML model does not have to make right predictions for the right reasons, so that the explanations obtained by analyzing model structures can be certainly unreliable. We then present an in-depth explanation in Section 3.6, where the results of all four experiments are connected.

Please find our responses to each of your comments below. More details about the major changes can be found in the manuscript document with major changes explained. Please let us know if you have further suggestions. Thank you again for reviewing our paper.

**(General comments)**

*I expect the Authors to further improve the article in order to overcome my main concerns:*

*1. First, I believe that a Machine Learning-based approach is less suitable for addressing a SuDS problem than a physically based systemic approach. If it is true that in some cases the geometric and hydraulic characteristics of these systems are not well known, it is even more true that the presence of experimental field data on inflows and outflows represents a rare exception. For this reason, a physically based modeling is in most cases to be preferred and allows to address a wider variety of problems, while in this case a model based on Machine Learning algorithms would be limited to the specific case. Authors should give more convincing reasons for the choice of approach.*

We fully agree with your point that ML methods are less suitable for modeling SuDS. In fact, we also mostly use process-based models, such as SWMM and GIFMod, in our SuDS-related studies; and we have written papers to improve these models. Please find below our justifications for conducting this research and the modifications made to the manuscript.

In this paper, we apply ML methods to SuDS studies based on the following considerations.

1. ML as a solution to benchmarking process-based models in SuDS-related studies. In our own research, we sometimes encounter SuDS sites that are so uniquely designed that we have to significantly twist the model presentations in the commonly used process-based models. However, we do not know whether such adjustments have led to satisfactory prediction accuracies if a "satisfactory" accuracy level is not defined. This study demonstrates that ML models can be set up quickly with little effort so that the prediction accuracies derived using ML methods can be used as a reference.

2. SuDS catchments as representations of small urban catchments. We certainly agree that monitoring data for SuDS are limited, and if such data are available then the other types of field measurements normally should also be available. We intended to use SuDS catchments as representations of small-scale urban catchments, and for which more data are available. For instance, the United States Geological Survey (USGS) has monitored many small streams near cities and the data are publicly available. The Shayler Crossing Watershed (SHC) studied

in the paper is in fact a small urban catchment (around 1 km$^2$) with SuDS features, which is of a much larger scale than a SuDS site.

3. It is easier to verify the physical realism of the inferred physical processes in small catchments compared to large natural catchments. For example, runoffs of a small catchment are expected to be affected mostly by recent rainfalls. However, in large natural catchments, it is difficult to identify the forcing factors that contribute the most to runoff generation, as the snow melting and baseflow processes that are related to the long-term climate patterns may be the dominant processes.

We made the following changes in the updated manuscript to make the justifications for conducting this research clearer.

1. We explained that ML models should be used as a reference to evaluate process-based models so that ML models should not be considered as a replacement to process-based models.

   *"The resulting statistical models may be adopted for solving various prediction tasks and used as references to assess the prediction accuracy of process-based models."* (line 51 to 52)

   *"Thus, in future SuDS studies, it can be useful to quickly train some ML models based on available data and used them as a reference to evaluate the prediction accuracies of process-based models."* (line 421 to 423)

   *"ML models can be set up relatively easily provided that observation data of the variables of interest are available and thus are recommended to be used as a reference to evaluate process-based models."* (line 585 to 587)

2. We commented that the SHC study is conducted to test how well the proposed method performed for small urban catchments.

   *"SHC represents a typical residential area in the U.S. and is thus selected to test the applicability of the proposed ML methods in modeling small urban catchments."* (line 326 to 327)

   *"The proposed ML model training methods can be potentially extended to study other small scale urban catchments that have similar configurations to SHC."* (line 423 to 424)

3. We explained that it can sometimes be difficult to come up with expected catchment hydrological behaviors, which are used for assessing the consistency of the inferred processes.

   *"The inability to draw a definitive conclusion can also be caused by the lack of knowledge of the processes being modeled. For instance, it is difficult to identify the expected hydrological behaviors for ungauged natural catchments." (line 305 to 306)*

Finally, we also brought up the issues of trustworthiness of ML models in the introduction section, so that we remind the readers that we should be critical against ML models as they are opaque.

   *"Most of the current hydrology literature uses post-hoc explainability techniques to test whether an ML model makes right predictions for the right reasons, where a model is*

*generally considered more trustworthy if it can generate predictions in a way that is consistent with our knowledge of the system being modeled."* (line 91 to 93)

*2. The novelty of the work is not relevant from a methodological point of view. The XGBoost algorithm is widely used in literature, as well as the other tools (Nested cross validation, Bayesian optimization, etc.) used in modeling. Authors should better highlight in the introduction section why this work would represent a significant upgrade over existing literature, worthy of publication in HESS.*

Thank you very much for raising the concerns about the novelty of this paper. We fully agree that the application of XGBoost and other ML tools is not innovative. The innovative part of this paper is on evaluating the physical realism of the hydrological processes inferred from ML modeling results, i.e., we examined whether ML models make the right prediction for the right reasons. However, in previous submissions, we emphasized too much on prediction accuracy estimation and model training method, and the content on the physical realism estimation is hard to find due to the lack of logical connections between different results. The previous submissions are more like "engineering" papers rather than research papers because the interesting findings are reported along with the ML prediction accuracy evaluation results without a proper definition of the research questions.

To address the issues mentioned above, we rewrote most parts of the paper, with a shift in focus from building accurate prediction models to testing the physical realism of inferred hydrological processes. We present below the innovative points and main findings, and our attempt to improve readability according to the updated research objective is presented in our response to your next comment.

1. A framework to assess the physical realism of the inferred physical processes is defined in the methods section, which emphasizes the importance of assessing the reliability of the inferences.
2. A high-level overview of the activities of inferring physical processes being modeled from ML modeling results is shown in Section 3.6.
3. We showed both conceptually and empirically that large uncertainty exists in every step of the inference processes. However, many current studies overlooked these uncertainties, and the inferred processes are regarded as "semi-truths" or findings. We consider that it is important to broadcast this information to hydrological communities.
4. We showed empirically that a model's prediction accuracy is not necessarily correlated with its ability to provide consistent explanations to the physical processes it models. That is, an ML model can make right predictions for the wrong reasons. Thus, we should not consider explanations produced by more accurate models as closer to reality.

*3. The article is very long, and in some places still unclear. It could certainly be shortened without compromising its contents. This is the most important aspect on which Authors should focus their efforts in order to obtain a more concise and clear manuscript.*

Thank you very much for the comment on the readability of this paper. We consider the lack of readability was mainly caused by an improperly defined research question in the previous

submissions. We were too critical about process-based models in the introduction section, and the goal is to build more accurate ML models. However, in the results section, we reported that ML models can generate inconsistent explanations to the physical processes, which lower their credibility and are confusing results. However, too much effort was spent on introducing an ML model training method and assessing the prediction accuracy, and we did not clearly explain why and how to assess the consistency of the explanations.

To address this issue, we redefined the objective as,

*"Therefore, in hydrological studies, it is meaningful to ask whether post-hoc explainability techniques and ML models can provide physically plausible explanations to the processes of the system being modeled and whether a model's abilities to provide accurate predictions and plausible explanations are correlated."* (line 103 to 105)

To accommodate the updated objective to improve readability, we made the following changes to the manuscript.

1. The introduction and methods sections have been modified according to the updated objective. We reduce the content on the shortcomings of process-based models, introduction to detailed ML methods, and added more explanations on testing the physical realism of the explanations of ML models.
2. We removed 3,000 words from the manuscript, which is about 25% of the main content of the previous submission. The writing is more precise in the updated manuscript.
3. We removed 5 equations and some detailed introduction to various ML methods.
4. The detailed introduction to ML model explanation methods has been removed, as they are not essential for understanding the study results.
5. Three graphical illustrations are added. Figure 1 shows briefly the raw input, the features, and the output of ML models, so the readers can more easily relate the equations to hydrological modeling. Figure 3 shows concrete examples of the processes of explaining the basis of model prediction, the process of inferring physical processes being model, and procedures to evaluating physical realism of the inferred processes. The concrete examples are discussed when each process is introduced in the method sections. We believe this can help the readers understand the methods better. Figure 10 is added to illustrate the process of inferring physical processes from ML modeling results from a higher level. To facilitate understanding, we use a cake metaphor, where we compare the ML explanation-based inference process to the process of identifying the ingredients used in a cake. This metaphor allows easier explanations of the uncertainties involved.
6. We removed much content on testing ML models' prediction accuracies and collectively analyzed the results of the two catchments. (1) In the previous submission, we used 4 similar feature engineering methods. And in the updated manuscript only one method is presented, as they are overall similar. (2) The results on evaluating whether overfits occur were removed. (3) The results on estimating the "water age" of stormwater runoffs were removed. (4) The physical realism of both study sites is now examined together, in the same figure, and described using the same paragraph.
7. We shortened the conclusion section significantly; the new conclusions are presented using bulletin points for clarity. Many non-essential findings are omitted.

**(Specific comments)**

Furthermore, I would ask the Authors to consider the following additional comments:

4. P3 L67 - "Machine learning methods, also referred to as data-driven modeling, predictive modeling, and statistical learning": These terms are not strictly equivalent.

Thank you for pointing the issue related to terminology. In the updated manuscript, we state that "Terms that are closely related to ML include data-driven modeling, predictive modeling, and statistical learning." (line 54), which are more precise than the original statement.

5. P4 L126 – Section 2.1.1 Local and global methods: This section is very basic and not essential for the manuscript. Sections 2.1.1 and 2.1.2 should be merged and shortened.

We have removed section 2.1.1, which is a basic introduction to ML explanation methods. Section 2.1.2, which is on the SHAP method, has now been shortened by 45% (from 900 words to 400 words). For more details, please see our responses to your comment #3.

6. P14 L183 – "The water levels are converted into discharge measurements using stage discharge rating curves": How were the curves obtained? Do they come from an appropriate calibration?

We removed this sentence on water level-discharge conversion from the updated manuscript for more concise presentations. The conversion was performed by USGS. In the updated manuscript we just state "The rainfall-discharge data collected between 2010 and 2013 by USGS are used in this study." (line 353 to 354)

7. P16 L426 – "The feature engineering and XGBoost hyperparameters are automatically optimized using the Bayesian optimization": Authors should report the optimal values of the hyperparameters, possibly in a table.

Thank you for your suggestion. We added Table 1 to report the range of the considered values. However, the optimal values are not shown because more than 400 optimal models are trained in this study. The optimal hyperparameters and the intermediate modeling results (together with all the source code) can be found in the link provided in the Code availability section.

8. P18 L486 – Nash-Sutcliffe efficiency coefficient and coefficient of determination $R^2$ are very similar metrics, it is not useful to consider both.

Thank you for sharing your opinion regarding the model performance metrics. $R^2$ is used because it is a well-known metric used in statistics and ML, so readers from different fields may look for R2. In some recent hydrological papers, both NSE and $R^2$ are used, e.g., in Sahour et al. (2020) and Yang et al. (2020).

However, we do agree that discussing two similar metrics is redundant. Therefore, in later experiments, such as experiment #4 and Figure 9, only NSE metrics are used. Also, in the abstract and conclusion section, only NSE metrics are reported.

9. P28 L668 – Conclusions: This section should be much more concise and effective in summarizing the main findings of the study.

Thank you for your comments. We reduced 500 words (i.e., a 75% reduction) from the conclusion section. The main conclusions are now presented as bulletin points for clearer presentations.

**References**

Sahour, H., Gholami, V. and Vazifedan, M.: A comparative analysis of statistical and machine learning techniques for mapping the spatial distribution of groundwater salinity in a coastal aquifer, J. Hydrol., 591, 125321, doi:10.1016/j.jhydrol.2020.125321, 2020.

Yang, W., Yang, H. and Yang, D.: Classifying floods by quantifying driver contributions in the Eastern Monsoon Region of China, J. Hydrol., 585, 124767, doi:10.1016/j.jhydrol.2020.124767, 2020.